METHODS AND RESOURCES

# A high-throughput behavioral screening platform for measuring chemotaxis by *C. elegans*

Emily Fryer[1,2☉], Sujay Guha[1☉], Lucero E. Rogel-Hernandez[1☉], Theresa Logan-Garbisch[1,3☉], Hodan Farah[1,2], Ehsan Rezaei[1], Iris N. Mollhoff[4], Adam L. Nekimken[1,5], Angela Xu[2], Lara Selin Seyahi[1,2], Sylvia Fechner[1], Shaul Druckmann[6], Thomas R. Clandinin[6], Seung Y. Rhee[2¤], Miriam B. Goodman [1]*

**1** Department of Molecular and Cellular Physiology, Stanford University, Stanford, California, United States of America, **2** Department of Plant Biology, Carnegie Institution for Science, Stanford, California, United States of America, **3** Neurosciences Graduate Program, Stanford University, Stanford, California, United States of America, **4** Department of Biology, Stanford University, Stanford, California, United States of America, **5** Department of Mechanical Engineering, Stanford University, Stanford, California, United States of America, **6** Department of Neurobiology, Stanford University, Stanford, California, United States of America

☉ These authors contributed equally to this work.
¤ Current address: Plant Resilience Institute, Departments of Biochemistry and Molecular Biology, Plant Biology, and Plant, Soil and Microbial Sciences, Michigan State University, Lansing, Michigan, United States of America
* mbgoodmn@stanford.edu

**Data Availability Statement:** All relevant data are within the paper and its supporting information, and from the Stanford Digital Repository (SDR): https://doi.org/10.25740/rh109rs1058. The SDR

## Abstract

Throughout history, humans have relied on plants as a source of medication, flavoring, and food. Plants synthesize large chemical libraries and release many of these compounds into the rhizosphere and atmosphere where they affect animal and microbe behavior. To survive, nematodes must have evolved the sensory capacity to distinguish plant-made small molecules (SMs) that are harmful and must be avoided from those that are beneficial and should be sought. This ability to classify chemical cues as a function of their value is fundamental to olfaction and represents a capacity shared by many animals, including humans. Here, we present an efficient platform based on multiwell plates, liquid handling instrumentation, inexpensive optical scanners, and bespoke software that can efficiently determine the valence (attraction or repulsion) of single SMs in the model nematode, *Caenorhabditis elegans*. Using this integrated hardware-wetware-software platform, we screened 90 plant SMs and identified 37 that attracted or repelled wild-type animals but had no effect on mutants defective in chemosensory transduction. Genetic dissection indicates that for at least 10 of these SMs, response valence emerges from the integration of opposing signals, arguing that olfactory valence is often determined by integrating chemosensory signals over multiple lines of information. This study establishes that *C. elegans* is an effective discovery engine for determining chemotaxis valence and for identifying natural products detected by the chemosensory nervous system.

repository holds all raw image files and the metrics derived from these images. The Neuroplant OWL (https://github.com/Neuroplant1062 Resources/ Neuroplant-OWL; DOI: 10.5281/zenodo.11122497) is custom software converting images to *.csv files holding the centroids of all detected worms. Code used to clean the data, perform statistical analysis, and generate the figures is available at https:// github.com/Neuroplant1065 Resources/ Neuroplant-DataAnalysis (DOI: 10.5281/zenodo. 11122497).

**Funding:** This work was supported by the Wu Tsai Neuroscience Institutes at Stanford via the Big Ideas and Research Accelerator programs (to MBG, TRC, & SYR) and a NeURO fellowship (to HF), the National Institutes of Health (R35NS105092 to MBG; F31NS100318 to ALN; T32GM113854 to LR-H), the National Science Foundation (IOS-1546838 to SYR), Chan-Zuckerberg BioHub (Investigatorship to TRC), and Stanford University (Rise Seed Grant to LR-H, SG; BioX Interdisciplinary Fellowship to LR-H). The funders had no role in study design, data collection and analysis, decision to publish, or preparation of the manuscript.

**Competing interests:** The authors have declared that no competing interests exist.

**Abbreviations:** CBI, chemotaxis buffer + iodixanol; CSN, chemosensory neuron; DMSO, dimethyl sulfoxide; DPI, dots per inch; GPCR, G-protein coupled receptor; NGM, nematode growth medium; OWL, Our Worm Locator; SM, small molecule; TRP, transient receptor potential.

## Introduction

Odors and other chemical cues shape behaviors like feeding, mating, and the avoidance of predators and other hazards. Humans and other animals, including invertebrates, perceive attractive odors as pleasant and repellent ones as foul and reliably classify chemical cues according to this single dimension of valence [1–3]. This process starts when odor molecules bind to receptors expressed by specialized chemosensory neurons (CSNs). In mammalian and nematode CSNs, odors and pheromones are typically detected by G-protein coupled receptors (GPCRs), and GPCR activation is transduced into electrical signals via activation of adenylate cyclase and cyclic-nucleotide gated ion channels or phospholipase C and transient receptor potential (TRP) channels. How these molecular and cellular events culminate in similar behaviors (approach or withdrawal) across phyla is an incompletely understood and fundamental problem in neuroscience.

The roundworm *Caenorhabitis elegans* has provided compelling insights into the genetic, molecular, and neural basis of chemosensation for 5 decades (reviewed in [4]). A primary strategy worms use to accumulate near attractants is to suppress turns (pirouettes) and to increase forward run duration when moving up a chemical gradient [5]. The converse strategy underpins the avoidance of repellents [6]. They also bias their heading during runs (weathervane mechanism) [7] and modulate their speed in chemical gradients [8]. Collectively, these strategies make it possible to monitor chemotaxis by observing the position of groups of animals following timed exposure to spatial chemical gradients. In hermaphrodites, chemotaxis behavior depends on signaling by one or more of the worm's 32 CSNs, organized into 16 classes of neuron pairs [4]. Thirteen classes innervate anterior sensilla, and 3 classes innervate posterior sensilla. Roughly 3 dozen organic chemicals and salts are thus far known to elicit chemotaxis. Some individual classes of CSNs are associated with promoting attraction or repulsion (for instance, [9]), mirroring the single dimension of valence. While ample evidence links specific odorants to particular CSNs and the receptors they express, how the broader chemical space of odorants that a worm might encounter could interact with one or more receptors to produce either attraction or repulsion is incompletely understood.

With a genome encoding more than 1,300 GPCRs, including receptors for neurotransmitters, peptides, and proposed chemosensory receptors (reviewed in [10]), *C. elegans* has substantial capacity for chemical sensing. Each class of CSNs expresses a distinct ensemble of hundreds of GPCRs [11,12]. With the exception of mammalian olfactory receptor neurons [13], many mammalian cell types also express hundreds of GPCRs [14]. Chemosensory transduction by hundreds of GPCRs expressed in *C. elegans* CSNs is thought to converge on either TAX-4-dependent cyclic nucleotide-gated ion channels or OSM-9-dependent TRP channels. Among the anterior CSN pairs, 9 classes express TAX-4 [15], including 6 that also express OSM-9 [16]. Four CSN classes appear to express OSM-9 alone [16]. These expression patterns divide the 13 anterior CSNs into 3 groups (3 TAX-4 only, 4 OSM-9 only, and 6 TAX-4 and OSM-9), all of which use one or both ion channels as key effectors for chemosensory transduction.

In the wild, feeding and reproducing stages of *C. elegans* are found across the globe in decomposing plant matter [17–19] and must, therefore, navigate complex environments that contain a wealth of plant-derived secondary metabolites and other small molecules (SMs). It is estimated plants make at least 200,000 chemically distinct SMs and that many of these compounds are released into the environment where they affect animal and microbial behavior [20]. Thus, plant SMs are an important component of the natural environment of *C. elegans* and are very likely to be ethologically relevant chemotactic cues. In the laboratory, it is common to monitor *C. elegans* chemotaxis by observing the position of groups of animals

following timed exposure to spatial chemical gradients (for instance, [21]; see also reviews [4,22]). This artisanal method is not well suited for screening chemical libraries, however. Inspired by efforts to create semiautomated methods for measuring *C. elegans* life span [23] and feeding behaviors [24], we developed a chemotaxis platform and integrated analytic workflow compatible with testing chemical libraries for their ability to attract or repel *C. elegans*. Our approach integrates hardware, wetware, and software and supports performing chemotaxis assays at scale. Although this platform is compatible with any chemical library, we opted to screen plant SMs for their ability to evoke *C. elegans* chemotaxis. This choice is inspired by the interaction between plants and nematodes in natural environments and the idea that such a coevolution-inspired approach can deepen understanding of interspecies chemical cues and animal behavior.

By screening a curated library of 90 plant SMs and 6 reference conditions, we found a total of 37 SMs that evoked chemotaxis in wild-type *C. elegans*, but not anosmic mutants lacking *tax-4* and *osm-9*. Most of these chemoactive compounds (27 of 37) were attractants, and only 10 were repellents. A similar enrichment of attractants is also seen in prior studies of *C. elegans* chemotaxis [21]. Taking advantage of the scale of our approach, we dissected the dependence of these responses on perturbations of *tax-4* or *osm-9* and discovered that while a handful of odorants were dependent on a single transduction pathway, most were dependent on both. Strikingly, loss of either *tax-4* or *osm-9* function reversed the response valence of 10 compounds. This finding implies that the response valence exhibited in wild-type animals reflects integration of signaling from multiple CSNs and/or receptors. More broadly, these results suggest that many SMs engage receptors expressed in multiple sensory neuron types and that behavioral valence emerges from integration of signals across multiple CSNs. These data demonstrate the value of our high-throughput behavioral screening approach for characterizing diverse chemical libraries, reveal that plant-derived SMs are salient chemical cues for *C. elegans*, and set the stage for using phenotypic assays to discover novel actuators of the nervous system and their cognate receptors.

## Methods

### Custom chemical library curation

We assembled a custom library of 94 compounds and 2 null reference conditions (DMSO: DMSO and DMSO:water). To link our findings to prior studies [4], we included 2 compounds known to attract (isoamyl alcohol, diacetyl) and 2 known to repel (2-nonanone, 1-octanol) wild-type *C. elegans*. The other 90 compounds were SMs synthesized by plants, soluble in DMSO, and purchased from commercial suppliers (S1 Table). We used anhydrous DMSO to dissolve all compounds and limited freeze–thaw cycles to 3 or fewer. They were selected based upon a search of the published literature for SMs that attract or repel animals that consume plants and/or are known to induce physical effects on animals. We expanded the set by searching for SMs that were chemically similar to an initial set of compounds or synthesized in the same biosynthetic pathway as these SMs. The library includes SMs made by plants used in medicine, human foods, or human rituals, such as camphor [25], salvinorin A and its propionate analog [26], and sinomenine hydrochloride [27]. The library also includes 3 SM pairs that map to the same compound according to the CAS registration number but have different common names and were purchased from different suppliers. For this reason, the SM pairs provide a window in reproducibility. These SM pairs are CAS No. 496-16-2– 2,3-dihydrobenzofuran and coumaran; CAS No. 106-22-9—citronellol and β-citronellol; CAS No. 474-58-8—daucosterol and sitogluside.

## Chemical reagents

The chemical library was sourced as indicated in S1 Table. Other chemical reagents were obtained from Sigma-Aldrich.

## C. elegans strains

We used 4 *C. elegans* strains in this study:

1. wild-type [N2 (Bristol), [RRID:WB-STRAIN:WBStrain00000001];

2. PR678 *tax-4(p678)* III [RRID:WB-STRAIN:WBStrain00030785];

3. CX10 *osm-9(ky10)* IV [RRID:WB-STRAIN:WBStrain00005214];

4. GN1077 *tax-4(p678)* III; *osm-9(ky10)* IV.

For the purposes of this study, N2 (Bristol) was the wild-type, *tax-4(p678)* and *osm-9(ky10)* are null alleles, and were derived in the N2 background. We made GN1077 by crossing GN1065 *osm-9(ky10)* IV*; pat-2(pg125[pat-2*::*wrmScarlet)* III with GN1076 *tax-4(p678)* III*; oxTi915 [eft-3p*::*GFP*::*2xNLS]* IV and selecting nonfluorescent progeny as candidate *tax-4; osm-9* double mutants. The final double mutant was verified by PCR and sequencing using the following primers for *osm-9* (Forward -GCAGAAGAGAAACTCCTCAC; Reverse -CCACCTTCATAATCTCCAGC) and *tax-4* (Forward -CCAATGGAATTGGCTCTCCTC; Reverse -CATCCCAAGTCAGGATACTG).

## *C. elegans* husbandry

We maintained *C. elegans* in 10-cm plates (Fisher Scientific, 229695) on nematode growth medium (NGM) seeded with OP50 *E. coli* and generated age-synchronized cohorts of young adults suitable for behavioral testing, using standard methods [28]. We thawed animals from frozen stocks prior to each round of screening and maintained them on OP50-seeded 10-cm NGM growth plates for several generations prior to using them for screening. The procedure for age-synchronization was as follows: (1) using sterile, filtered, osmotically purified water, wash worms from growth plates into 15-mL conical tube; (2) concentrate worms by centrifugation (1 minute, 4,000 RPM, Thermo Scientific Sorvall Legend X1R), discard the supernatant, and distribute the pellet in approximately 250 µL pellets into 15-mL tubes; (3) resuspend pellets in water (4 mL) and add household bleach (1 mL) and 5M KOH (0.5 mL), vortex, and incubate until adult worms disintegrate and eggs are released (5 to 10 minutes); (4) concentrate eggs by centrifugation (1 minute, 4,000 RPM) and discard the supernatant; (5) wash in water (10 mL) and concentrate by centrifugation (1 minute, 4,000 RPM), 4 times; and (6) resuspend egg pellets in water (2 mL) and deliver 1,200 to 1,800 embryos onto OP50-seeded, 10-cm NGM growth plates. Animals were incubated at 20˚C and reached adulthood in approximately 3 days; only well-fed cohorts were used for behavioral testing.

## Chemotaxis assays

We conducted end-point assays of populations of synchronized, young adult wild-type and mutant *C. elegans*. Our implementation involves novel behavioral arenas (4 per assay plate), methods for linking the chemical library format to assay plates, strategies for dispensing worms using automated liquid handling equipment, and humidity-controlled environments for running assays (S1 Fig). For each strain, we collected and analyzed the data from at least 3 biological replicates, which consisted of independently prepared, age-synchronized worms tested on different days. This enabled us to detect systematic variations in husbandry or assay

conditions, if present. The data were pooled across biological replicates since no variation was observed. We masked the identity of test compounds and *C. elegans* genotypes during all experiments, which were performed by a team of 2 investigators.

**Behavioral arenas.** We used thin foam to define assay arenas because it is hydrophobic, non-absorbent, and easy to cut with precision and reproducibility [29] with a computer-controlled cutting machine (Cricut Maker and Cricut Maker3, Cricut). We used thin sheets of EVA foam (Cleverbrand, 9" × 12" × 1/16" or BetterOfficeProducts, 9" × 12" × 1/12"). The precise dimensions of each insert are shown in Fig 1B, and we cut several inserts from a single 9" × 12" foam sheet. Notably, the apex-to-apex distance (6.8 cm) is comparable to the 6-cm distance between test and reference chemicals used in classical chemotaxis assays [22]. We filled assay lanes with gellan gum (Gelrite, Research Products International, G35020-100.0) instead of agar, floating precut foam inserts on top of the molten media so that it formed a worm-proof seal as the media solidified at room temperature. We sealed assay plates in plastic wrap and stored them at 4°C for up to 14 days prior to use. We selected gellan gum because of its superior optical clarity (Fig 1C) and settled on 2.5% (w/v) concentration as a practical balance between cost, stiffness, and clarity. We dissolved gellan gum (2.5% w/v) in ddH$_2$O and heated it above 75°C by autoclaving. Chemotaxis buffer [5 mM KPO$_4$, pH 6, supplemented with MgCl$_2$ (1 mM) and CaCl$_2$ (1 mM)], prepared as described in [30], was added when the media cooled to 60°C. Using serological pipettes, we added buffered, molten gellan gum (10 mL) to each assay lane and floated precut foam inserts (see below) on top of the molten media.

**Chemical gradient setup.** We arrayed our chemical library into 96-well microplates at a concentration of 20 mM in dimethyl sulfoxide (DMSO) for all compounds except the reference set. Attractive reference compounds (isoamyl alcohol, diacetyl) were diluted serially in DMSO to 1:1,000, 2-nonanone was diluted to 1:10, and 1-octanol and DMSO were added directly to the plates. These concentrations were drawn from the literature and take into account the observation that a single compound can elicit attraction or repulsion, depending on concentration [31–34]. We anticipated that a subset of our compounds might not be soluble at this concentration; indeed we observed precipitates for 17 compounds or 18% of the curated library (denoted with (p) in S1 Table). This fraction is comparable to the 6% to 19% of large chemical screening libraries reported to be insoluble in DMSO [35].

For all assays, compound identity was masked until after screening was completed. We used a variable-spacing multichannel pipette (Thermo E1-ClipTip 2–125 μL) to transfer 3.5 μL of each compound from the chemical library plate into assay plates (Nunc 4-well plates, Thermo Fisher, Cat # 267061). We used each of the lanes of a vented 4-well multiwell assay plate to create 4 two-dimensional behavioral arenas consisting of solid media and a custom-fabricated foam corral in each multiwell plate (S2 Fig). To reduce cross-talk and retain volatile chemicals within each lane, we inserted foam sheets (3.24 in × 4.92 in) into the lid of the assay plate. Test compounds were dispensed into one apex, and the solvent, DMSO, was dispensed into the opposite apex, both without added sodium azide. Assay orientation was standardized by delivering test compounds to the notched side of each arena (Fig 1A and 1B). Once loaded with test compounds and the solvent vehicle, we held assay plates at room temperature for 1 hour to establish a chemical gradient.

**Preparing worms for large-scale behavioral assays.** We generated synchronized populations of worms and collected them for behavioral assays as follows. First, we examined NGM agar growth plates for signs of starvation or microbial contamination and discarded plates with starved animals or visible contaminants. Next, we collected young adult worms in 2.5 mL of sterile ddH$_2$O, gently swirling the plate to dislodge worms from the agar surface. We transferred the worm slurry to a 15-mL conical tube, concentrated the animals in a centrifuge for 1 minute at 3,000 RPM, and washed the worm pellet 3 times with sterile ddH$_2$O to remove trace

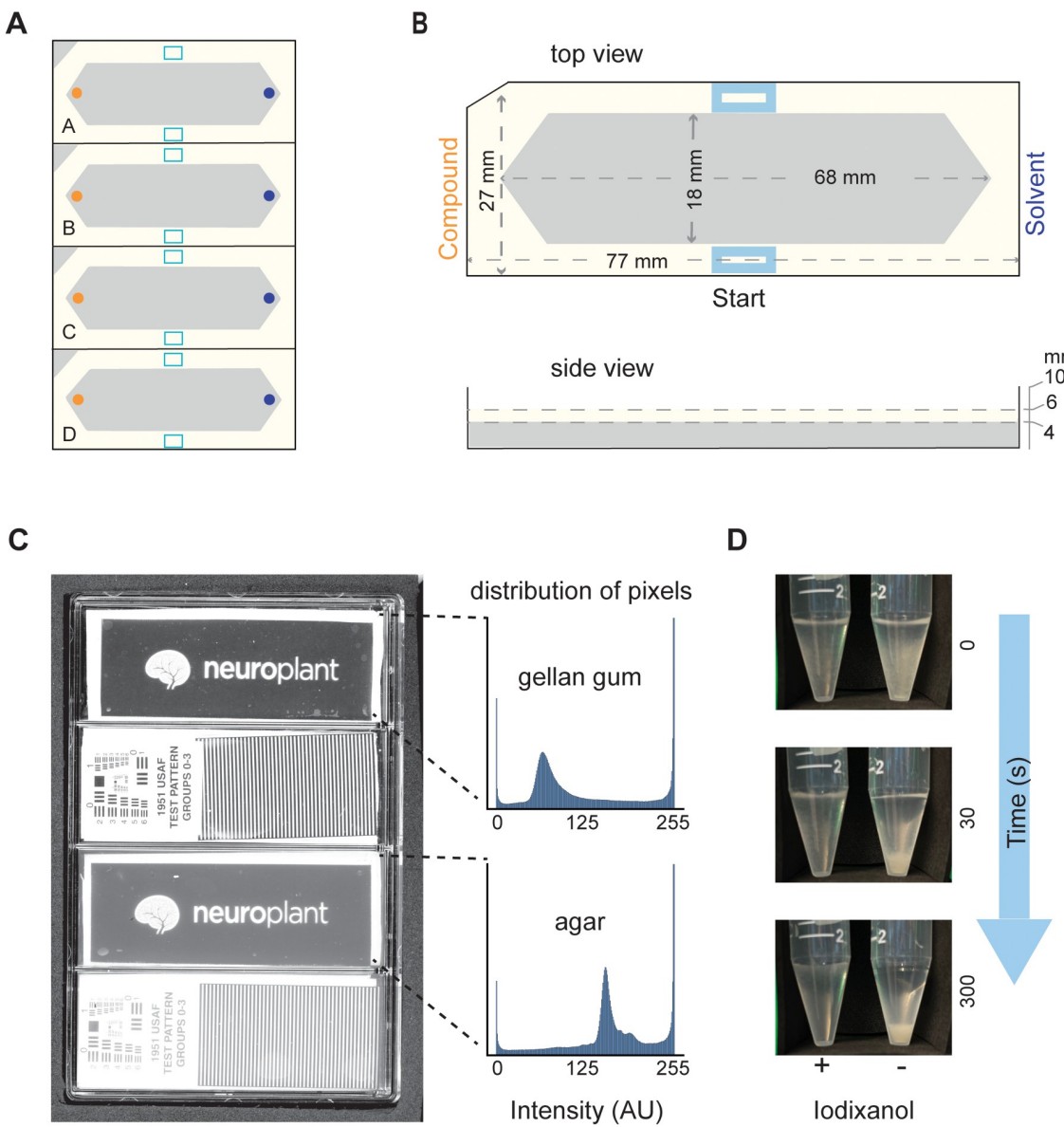

**Fig 1. Measures that enable increased throughput of population-based *C. elegans* chemotaxis assays. (A)** Schematic of a 4-lane assay plate (standard microtiter plate footprint) showing foam inserts. (**B**) Top and side view dimensions of a single foam insert. Panels (**A**) and (**B**) illustrate the assay starting zone (light blue), position of the test compound (side with notched corner, orange), and the reference or solvent (opposite, dark blue). (**C**) Image collected on a flatbed scanner of a single 4-well assay plate (left) containing Gelrite gellan gum (top 2 lanes) and agar (bottom 2 lanes). Transparent test patterns (Neuroplant logo, 1951 USAF test pattern) placed on the surface of the solid media are used to illustrate improved clarity for gellan gum compared to agar. Intensity histogram drawn from the image of the test pattern imaged through gellan gum (top) and agar (bottom). (**D**) Still images of a time lapse observation of worms suspended in chemotaxis buffer with (+, left) and without (−, right) Optiprep solution of iodixanol (7:3 chemotaxis buffer: Optiprep).

*E. coli* OP50. Washed pellets were resuspended in a 7:3 ratio of chemotaxis buffer (see above) and Optiprep (60% iodixanol in sterile ddH$_2$O w/v, Sigma-Aldrich, D-1556), resulting in a net dilution to 18% iodixanol w/v. As shown in Fig 1D, worms remain suspended in chemotaxis buffer + iodixanol (CBI), an effect that reduces the variation in the number of worms delivered using liquid handling instruments or by manual pipetting. Iodixanol is a nontoxic polymer used to tune the index of refraction for live imaging applications [36,37]. It is also used in

density gradient centrifugation applications [38], making it an ideal chemical tool to improve consistency of dispensing *C. elegans* in liquid. Finally, we resuspended an approximately 0.5 mL worm pellet in 3.5 mL of CBI to deliver approximately 250 worms/assay arena onto 12 assay plates.

**Liquid handling and worm dispensing.** To increase throughput and reduce trial-to-trial variation of the number of worms dispensed into each assay arena, we adapted a multimode reagent dispenser (Biotek, Multiflo) and plate stacker (Biotek Biostack 3) to automatically dispense worms suspended in CBI. In brief, we separated a single line near the center of an 8-channel cassette (10 uL #423526) and adjusted the Liquid Handling Control software (LHC 2.22.) to deliver worm-laden drops in the center of the assay arena. To achieve this goal with sufficient precision, we used the 1,536-well preset configuration in the LHC software to deliver a single droplet at the center of each of the 4 wells. Finally, we adjusted the flow rate and dispensing volumes to minimize splatter during dispensing events and droplet spread while the plates were in motion on the working surface of the liquid handler and plate stacker. Once the dispense cycle was completed, we flushed the line of any remaining worms by flowing 100% ethanol for 10 seconds, followed by ddH$_2$O for 20 seconds. Using this approach, we dispensed 100 to 450 worms into each arena (approximately 20 seconds per plate) and processed 12 plates in parallel for a total elapsed run time of approximately 250 seconds.

**Running the chemotaxis assay.** Once dispensed onto the assay plate, worms were retained in the liquid droplet. Thus, excess liquid needed to be removed to disperse animals and enable free movement. To achieve this goal, we placed absorbent PVA eye spears (BVI Ultracell -40400-8) on the center of the liquid droplet to withdraw as much liquid as possible by capillary action and used the fine point of the eye spear to disperse animals across the width of the assay arena, disrupting clumps of animals. Finally, assay plates prepared with chemical gradients and animals were transferred to a dry cabinet (Forspark, Cat.# FSDCBLK30) set at 31% relative humidity and allowed to move freely for 1 hour at room temperature (20 to 24˚C).

## Image capture

To efficiently capture images of the distribution of worms at the end of each chemotaxis assay, we used flatbed scanners (Epson, Perfection V600 Photo). We captured 8-bit grayscale images at 1,200 dpi, with both brightness and contrast set at 50, choosing these settings to maximize contrast and resolution of the worms. The scan-bed on this instrument was large enough to simultaneously scan 4 assay plates positioned on the scanner surface using a frame cut from a sheet of black foam (9" × 12" × 1/6", Cleverbrand Fun Foam, Black). The frame helped to map the 4 plates captured in a single image to their respective metadata and increased image contrast by setting consistent black levels. Each plate was scanned once and held in the scanning environment for approximately 2 to 3 minutes, during which time the temperature did not increase (before: 21.94 ± 0.08˚C, mean ± SD, *n* = 5; after: 21.85 ± 0.03˚C, *n* = 4; mean ± SD), measured every 30 seconds using LabJack Digit-TLH data logger). In addition, we adjusted the position of the scanner's camera lens to achieve a sharp image at the surface of the gellan gel media that formed the assay arena. Specifically, we used standardized, transparent resolution patterns (USAF, 1951 Test Patterns, Edmund Scientific, #38–710) placed in position mimicking the assay surface and adjusted the lens position to maximize image sharpness, as proposed [23]. Assay plates were too tall to fit inside the standard scanner lid, which we removed and then enclosed each scanner in a black plastic storage container (Sterilite, 65.4 cm L × 46.7 cm W × 18.1 cm H; S1 Fig). Collectively, these measures resulted in high-contrast images having a standardized layout. Sub-images of worms had sharp borders, indicating that animals were not likely to be moving during the scanning procedure.

## Image processing to locate worms

We transformed endpoint images of assay plates into arrays of worm positions in each assay arena using a custom Python (v 3.7.4) code base. The software, which we call OWL (Our Worm Locator) locates the centroid of animals in scanned images and is built upon scikit-image [39]. Raw 8-bit 1200 dpi grayscale Tagged Image File Format (*.tiff) files of the chemotaxis endpoint were read and converted into Numpy (v 1.16.4) arrays. We used Otsu's method [40] to determine a global thresholding value and generated a binary matrix of pixels for each image. All pixels with a pixel intensity greater than the thresholding value were set to white and all pixels less than the thresholding value were set to black. We then used the close function to repair defects that occurred during binarization. Contiguous groups of white pixels were labeled, and all labeled objects were stored in a Pandas dataframe. These data included centroid location $(x, y)$, object area, and bounding box values. The white foam inserts are the largest detectable objects within the image and were used to sort the data frame and assign well IDs based on their area and $(x,y)$ position in the image. Using the coordinates of the foam insert, we generated a mask that allowed us to dynamically divide image scans into images of each well. Objects in the cropped image were then relabeled and filtered to retain only those with an area greater than 50 pixels and less than 2,500 pixels. This range of values excluded small objects that were not worms (eggs, dust, etc.) as well as large clumps of worms but included small clumps of worms that were counted as single objects. Instead of attempting to estimate the number of worms in clumps [41], we sought to reduce their occurrence by manually dispersing animals across the width of the assay arena in the starting zone. The $(x, y)$ centroid coordinates of each identified worm-like object were exported as a comma-separated values (*.csv) file for each well and used to evaluate chemotaxis. To support users, we used PySimpleGUI to create a graphical user interface for OWL.

## Metadata and digital data management

For each round of screening, we established and maintained 2 types of data files (location, summary) and 1 metadata (plate ID, strain ID, compound ID), connecting each assay to the conditions in that particular trial using Python scripts (see Code availability). Each assay arena is associated with a location file and a summary file. The location file contains the $(x, y)$ coordinates (in pixel units) of all the worms detected in the arena. We linked each location file to its assay conditions using an automated file naming convention in which the file name contained the image ID, scanner slot number (location of the plate in the scanned image), and the well ID (location of the well within the plate). The summary file contains the total number of worms counted in the assay arena, the calculated chemotaxis index, and the distance between apices (in pixels) (3,041 ± 20, mean ± SD, $N$ = 311 arenas), test compound, worm strain, image ID, and plate ID. All raw and processed data files are stored in open-source file formats (*.tiff, *.csv) or as Google Sheets. Each image is assigned a unique image ID, linking the image to its respective metadata and image analysis results. Metadata are stored as Google Sheets and include assay date, experimenter, image ID, plate ID, scanner slot number, compound ID, strain ID, relative humidity, and temperature.

## Assessing the accuracy of image-based measures of chemotaxis behavior

We assessed OWL's accuracy by comparing human- and machine-analyzed images. First, we identified 3 cropped endpoint images for the reference conditions [isoamyl alcohol, 2-nonanone, 1-octanol, symmetric DMSO (DMSO:DMSO), and asymmetric DMSO (DMSO:water)] and 3 cropped endpoint images for diacetyl. In total, 19 images were identified for human scoring. Next, 2 people were assigned to score each cropped image using the same manual scoring

protocol, described as follows. Each image was loaded into FIJI [42] and human counters logged the location of individual worms using the "multipoint" selection tool. Once all worms were located and logged in an image, the human counter used the "Measure" function to return the $(x,y)$ coordinates (pixel) of all counted worms in the image and exported these data as a *.csv file.

We used 2 metrics to analyze OWL's performance: (1) total number of worms counted in an assay arena and (2) the mean position of worms within an assay arena. For both metrics, we used Pearson's correlation coefficient (computed by linregress in scipy.stats, version 1.7.1) to evaluate the similarity between human scorers and between each human and the OWL software. Mean worm positions were calculated using the mean module in the Python statistics package (v 3.7.4). Residuals were calculated and plotted for both analyses using the Seaborn (v. 0.9.0) residplot package. Finally, we generated kernel density estimation plots to compare the worm locations in each well identified by both human scorers and OWL using the Seaborn kdeplot package (v 0.9.0).

## Data and statistical analysis

Each assay arena is associated with a *.csv file of the $(x,y)$ pixel positions (in units of dots per inch or DPI) of worms detected in the endpoint image of the experimental arena. In this coordinate system, the $x$ axis extends along the chemical gradient and the $y$ axis indicates position across the width of the arena. We collected images at a resolution of 1,200 DPI (pixels/inch), converted units from pixels to millimeters, and repositioned the origin of the $x$ axis to the center of the arena as follows:

$$z = (-x + w) \times 25.4mm/1,200\ DPI$$

where $z$ = worm position along the $x$-coordinate in mm, $x$ = worm position along the $x$-coordinate in pixels, and $w$ = distance between the arena apices in pixels. Positive values of $z$ indicate positions closer to the test compound and negative values for $z$ indicate positions closer to the solvent reference. As shown schematically in S2 Fig (Steps 4 and 5), the total range for $z$ is −32.5 to +32.5 mm.

We established and maintained metadata sheets to link these datasets to the conditions of each assay (see below) and used these datasets to evaluate trial to trial variation, pool results across trials, and to determine the effect of test compounds on chemotaxis. Our analysis approach used the distribution of animals along the axis of the chemical gradient, which we designated as the $x$ axis in our coordinate system, to determine chemotactic responses. Conditions that resulted in roughly equal numbers of animals migrating toward each apex in the arena and an average worm position indistinguishable from zero were considered evidence of indifference to the chemical conditions in the arena. On the other hand, distributions biased toward or away from the test compound were classified as positive and negative chemotaxis, respectively. We also refer to these outcomes as attraction and repulsion, respectively. We used the x-coordinate to determine both mean worm position and chemotaxis index. Mean worm position is the average value of the x coordinate and the chemotaxis index is computed from $(p − q) / (p + q)$, where $p$ and $q$ are defined as follows. First, we divided the apex-to-apex distance of the assay arena into nine equal segments. Next, $p$ was defined as the total number of worms in the 4 regions on the side of the test compound and $q$ was defined as the total number of worms in the four regions on the opposite side. The remaining 1/9th of the arena is the starting zone, and, consistent with prior practice, animals present in this zone at the end of the assay were excluded from the calculation of chemotaxis index.

The strength of each putative chemotaxis response was determined using estimation plots [43–45] comparing worm position evoked by exposure to test compounds with those found

for 2 null reference conditions: symmetric solvent (DMSO:DMSO) and DMSO opposite water (DMSO:water). Effect sizes (difference of mean values, termed "mean difference") were determined via a bootstrapping approach implemented by the Dabest software library (v 0.3.1) [43]. This computation generates a range of likely values for the mean difference between each test condition and the null reference or control condition and reports the 95% confidence intervals of this value, resampling the experimental data 5,000 times with replacement. Cases in which the 95% confidence intervals of the mean difference include zero are statistically equivalent to a failure to reject the null hypothesis. Conversely, cases in which the 95% confidence interval of the mean difference excludes zero indicate that the null hypothesis can be rejected with a significance of at least $p < 0.05$ [45]. To account for spurious results that might arise from multiple comparisons, we converted 95% confidence intervals to exact $p$-values and applied a Benjamini–Hochberg correction [46].

To perform multiple comparisons between 2 bootstrapped effect sizes originating from the response of our different genotypes to a given test compound, we made use of a 2-factor approach akin to a two-way ANOVA [45]. This analysis was performed using the delta–delta ($\Delta\Delta$) package provided by Dabest [43]. $\Delta\Delta$ comparisons are computed by taking the difference between $\Delta_1$ and $\Delta_2$, where $\Delta_1$ is defined as the difference in the bootstrapped symmetric DMSO ($C$) mean differences between genotype 1 ($X_{G1}$, $C$) and a secondary genotype ($X_{G2}$, $C$) and $\Delta_2$ is defined as the difference in the bootstrapped mean differences between genotype 1 ($X_{G1}$, $T$) and the secondary genotype ($X_{G2}$, $T$), relative to a given test compound ($T$).

$$\Delta_1 = (X_{G1}, C) - (X_{G2}, C)$$
$$\Delta_2 = (X_{G1}, T) - (X_{G2}, C)$$
$$\Delta\Delta = \Delta_1 - \Delta_2$$

Additional statistical testing was performed using scipy.stats packages (v 1.7.1).

### Structured literature review

To evaluate the novelty of SMs in our chemical library as either attractants or repellants of *C. elegans* or other nematodes, we designed and performed a structured search of the PubMed and Web of Science (WOS) databases on the subset of SMs we identified as either attractants or repellants. The search terms consisted of compound name, CAS No., and species name (*C. elegans* or *Caenorhabditis elegans*) or compound name, CAS No., and "nematode NOT elegans" together with "chemotax*". Next, we excluded studies that used plant extracts or complex mixtures, studies in which worms were used as pathogen vectors, or transformed with human peptides. Finally, we eliminated duplicates, generating a set of 61 unique publications.

### Code availability

We developed OWL and the OWL GUI software in Python version 3.10 and used Anaconda (v 2020.02) to set up a virtual environment that contains all of the Python packages and versions necessary to run these tools. The full codebase is publicly available in a Github repository, https://github.com/Neuroplant-Resources, and includes a *.yml file to define package and version information (NP_conda_env.yml file).

### Results

This work harnesses chemical communication between plants and nematodes [47–49] to identify SMs that are detected by the chemosensory nervous system. Our approach relies on testing SMs synthesized by plants for their ability to either attract or repel the model

roundworm, *C. elegans*. This behavior, known as chemotaxis, has at least 2 advantages for the purposes of identifying chemical cues detected by neurons. First, because animals are not immersed in test chemicals, there is little, if any, risk of lethality. Second, all putative receptors expressed by the 32 CSNs are tested in parallel. Each class of CSNs expresses a distinct ensemble of ion channels and receptors [10]. The data available from neuron-specific and single-neuron RNASeq datasets [12] and promoter fusions [11,50] indicate that a single CSN expresses approximately 100 GPCRs and 3 to 5 receptor guanylate cyclases and that no 2 classes of CSNs express identical subsets of either class of membrane receptors. Thus, by working with a defined sensorimotor behavior of the whole animal, we test as many as 1,000 putative receptors for plant SMs without building libraries of cells expressing putative receptors or establishing in vitro assays of their function.

## A 4-lane highway for nematode chemotaxis assays

We followed a rapid prototyping, design-build-test approach to retool classical laboratory assays for *C. elegans* chemotaxis. Our prototyping cycles were guided by these design rules: (1) minimize manual handling; (2) use uniform behavioral arenas; (3) use common scientific or consumer equipment; (4) automate analysis; and (5) integrate data acquisition and management. Classical *C. elegans* chemotaxis assays are often performed on round (6-cm or 10-cm diameter) agar plates bearing a chemical gradient created by a small volume of test compound at the edge of one side of the plate and the relevant vehicle on the opposite side (reviewed in [4]). Animals are dispensed into the center of the assay plate and allowed to move freely for a defined time. Following the assay, the number of animals on the compound and solvent sides are counted manually, and these counts are used to compute a chemotaxis index that has a value of +1 for ideal attractants, −1 for ideal repellents, and 0 for compounds that are not chemoactive. This chemotaxis assay is simple, widely used, and reduces a complex behavior (chemotaxis) to a single endpoint metric, but its throughput is limited.

Based on our goals and design rules, we selected standard multiwell plates with 4 lanes for behavioral arenas (Fig 1A). To further standardize assay arenas, we fabricated foam inserts and floated them on top of optically clear solid media (gellan gum) deposited in each lane (Methods). The foam's hydrophobic surface retains animals within the arena, and its shape standardizes the placement of both animals and compounds on the arena surface (Fig 1A and 1B). These choices allow for a workflow based on standard instrumentation compatible with multiwell plates. We exploited this feature by using a liquid handler and plate stacker to dispense worms onto assay plates. The liquid handler not only dispenses worms onto 12 plates (48 assay arenas) in approximately 4 minutes but also dramatically increases the repeatability and accuracy of the number of worms dispensed (coefficient of variation, $CV = 0.259$) compared to manual pipetting ($CV = 0.553$).

Worms do not stay suspended in conventional buffers, leading to systematic variations in the number of worms dispensed in liquid. We counteracted this effect using iodixanol, a non-toxic polymer, to adjust buffer density so that *C. elegans* are neutrally buoyant in solution. After 30 seconds, *C. elegans* animals in standard buffer form a visible pellet, but animals in iodixanol buffer remain suspended (Fig 1D). This effect reduces variability in dispensing animals and could be extended to other workflows, including those that rely on manual pipetting. The dispensing liquid must be dispersed before animals crawl freely on the gel surface. At present, this step is performed manually using lint-free, absorbent eye spears to withdraw excess liquid and to disperse animals across the width of the behavioral arena. Collectively, these maneuvers accelerate and improve chemotaxis assay reliability.

## Iterative improvements in imaging and automated chemotaxis measurements

We adapted a consumer flatbed scanner to rapidly image assay plates at high contrast and developed an image processing pipeline, Our Worm Locator (OWL), for detecting worm positions. Our prototyping cycle identified 4 modifications that were instrumental in reaching this goal. First, we replaced agar with gellan gum because agar lacks the optical clarity needed to achieve high contrast images (Fig 1C). Gellan gum is a natural heteropolysaccharide purified from the bacterium *Sphingomonas elodea* [51], which can be cast into stable, solid gels similar to agar. Second, we modified the flatbed scanner to achieve sharp focus at the gellan gum surface (Methods). Third, we used custom foam inserts to standardize behavioral arenas, to improve image contrast, and to simplify downstream image processing. We programmed the cutting machine to mark the worm starting zone equidistant from the apices of the arena (Fig 1A and 1B). The apices define locations for spotting compounds and solvent controls, while the hydrophobic surface repels worms, retaining animals in the main arena. Fourth, we cut black craft foam to generate guides for consistent positioning of 4 assay plates on the scan bed. The scanner captures a full-field image of 4 assay plates in approximately 2 minutes, yielding a single image at near-uniform time point. Fast, endpoint imaging eliminated the need to include sodium azide to trap worms near test compounds and solvent, as is typical in classical assays [21,22]. Because of the sharp contrast and consistent positioning, our codebase efficiently and reliably demultiplexes scanner images into images of single-assay plates and each assay plate is demultiplexed into single-assay arenas (S2 Fig). Compared to the initial iteration of the design-build-test cycle, these actions generated a 16-fold increase in data collection efficiency and a 40-fold increase in image capture efficiency.

## Imaging processing pipeline to determine worm position

Borrowing imaging principles from software for tracking worm movement [52] and similar to other reports [53,54], OWL locates and logs the $(x, y)$ centroid position of all worms in our assay arenas. OWL removed multiple, significant barriers to scaling up chemotaxis assays that depend on manual counting, which is time-consuming and error-prone. The OWL software, by contrast, determines the locations of hundreds of worms from images collected at a single time and generates large, digital datasets that can be efficiently analyzed at any time. As part of our design-build-test cycle, we pooled data across 16 assays in which animals were exposed to solvent (DMSO) on both sides of the arena (Fig 2A) and used bootstrapping techniques to determine how the number of assays in a given arena affects the chemotaxis index. Across 4 *C. elegans* genotypes (wild-type, *tax-4*, *osm-9*, and *tax-4;osm-9*), we observed the mean chemotaxis index, but not the variance, was independent of the number of worms. As expected for a random or pseudorandom process, variance was inversely proportional to the total number of animals (Fig 2B). Because the variance reaches a steady minimum near 150 worms, we used this value as a quality control threshold—including assays with at least this many worms and excluding those with fewer worms.

## Platform performance and validation

To assess pipeline performance, we tested the response of the standard laboratory *C. elegans* strain (N2, Bristol) to 4 compounds with well-established chemotaxis phenotypes and to 4 null conditions, predicted to result in indifference. We selected 2 attractants, isoamyl alcohol and diacetyl [4], and 2 repellants, 2-nonanone, and 1-octanol [4]. The 4 null conditions were as follows: DMSO (DMSO:DMSO or symmetric DMSO); DMSO versus water (DMSO:water); DMSO

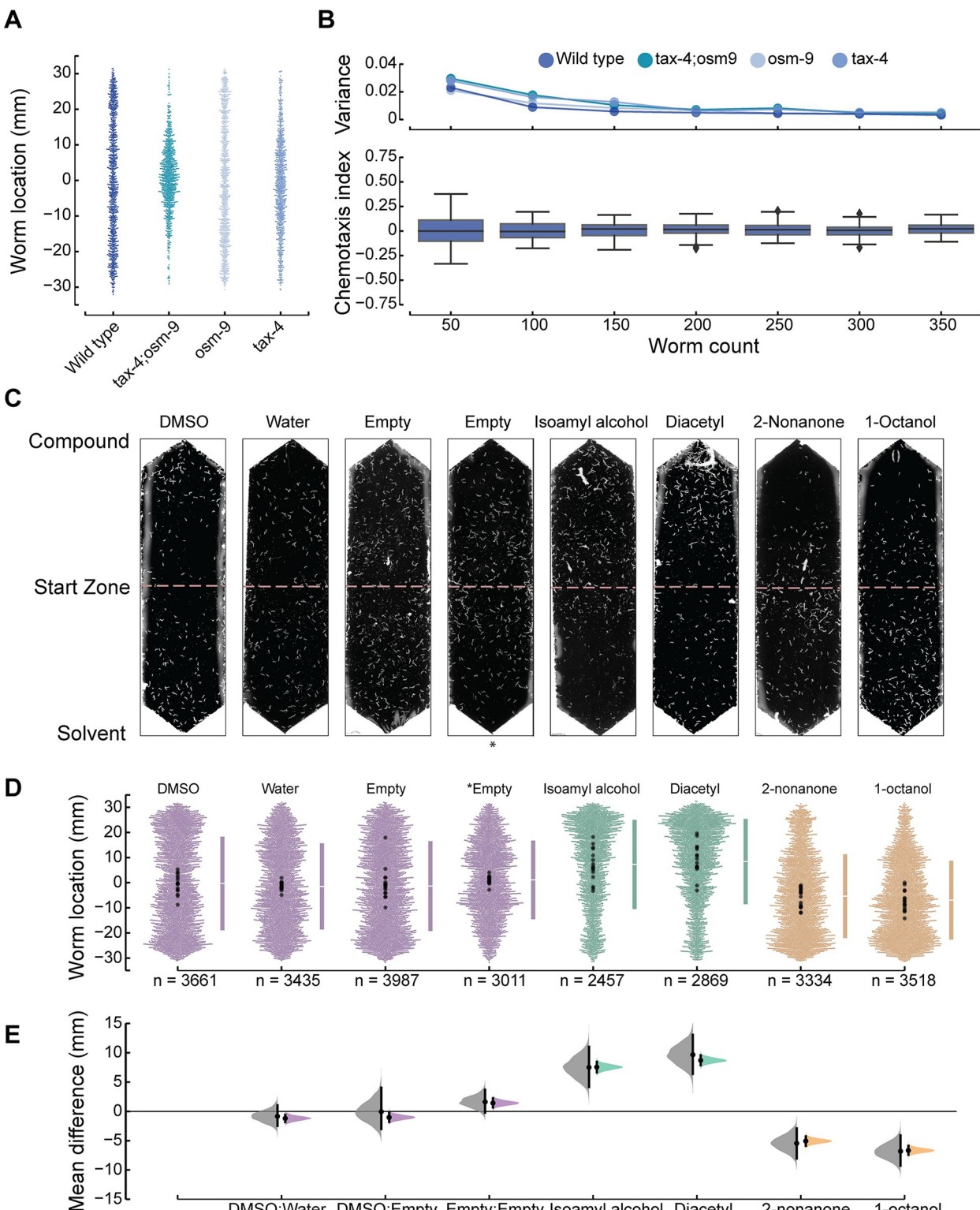

**Fig 2. Optimization and validation of chemotaxis performance and derivation of average position as a robust chemotaxis metric. (A)** Distribution of animals following exposure to symmetric DMSO. Each dot represents the *y* coordinate of a single animal of the indicated genotype, pooled across 3 biological replicates: wild type (N2), *tax-4(p678);osm-9(ky10)*, *osm-9(ky10)*, and *tax-4(p678)*. **(B)** Average (±SD) chemotaxis index for wild-type animals (bottom) and variance for the indicated genotypes (top) as a function of the number worms in an assay arena. The data are a bootstrap analysis of the data in panel (**A**) for sample sizes from 50 to 350 (increments of 50) animals. **(C)** Representative images of assay arenas

following exposure to (left to right): 4 null conditions, 2 known attractants, and 2 known repellents. DMSO is on the solvent (bottom) side, except for the empty condition denoted by an asterisk,*. (**D**) Swarm plots pooled across 16 technical replicates for each condition shown in panel (**C**). Bars to the right of each swarm show the ±1 standard deviation, with the gap between the bars indicating the mean worm position. Points are color-coded according to condition: null reference or control conditions (purple), attractants (green), repellents (gold). Larger points (black) are the mean worm location for individual replicates. (**E**) Effect size relative to the DMSO:DMSO null condition. Black bars and shaded areas show the difference of the mean values and the 95% confidence intervals for this value, bootstrapped from the data for each test condition. Leftward facing shaded areas (gray) represent the results considering each assay and rightward facing areas (colors) represent the results obtained by pooling across replicates. Mean differences [±95% CI] of the 16 assays are: DMSO:water, −0.84 [−2.67, 1.27]; DMSO:Empty, −0.06 [−2.99, 4.09]; Empty:Empty, 1.62 [−0.22, 3.69]; isoamyl alcohol, 7.50 [4.16, 11.00]; diacetyl, 9.65 [6.38, 13.05]; 2-nonanone, −5.45 [−8.05, −2.90]; 1-octanol, −6.80 [−9.24, −4.10]. Mean differences [±95% CI] of the pooled data are as follows: DMSO:water, −1.20 [−2.00, −0.40]; DMSO:Empty, −1.03 [−1.79, −0.27]; Empty: Empty, 1.43 [0.66, 2.21]; isoamyl alcohol, 7.55 [6.65, 8.45]; diacetyl, 8.70 [7.90, 9.55]; 2-nonanone, −5.07 [−5.89, −4.28]; 1-octanol, −6.66 [−7.40, −5.88]. Instances that exclude a mean difference of zero are considered bona fide responses compared to the null condition. Positive values indicate attraction (positive chemotaxis) and negative values indicate repulsion (negative chemotaxis). Data used to calculate these statistics and to generate these figures are reported in **S1 Data**.

versus empty (DMSO:empty); no compound added (empty:empty). We selected these conditions based on the use of DMSO as the solvent for all of our test SMs and to determine if animals were sensitive to this solvent. Fig 2C shows images of single assay arenas for the 4 null conditions and the 4 reference compounds. We plotted the position of every worm along the chemical gradient across 16 replicate assays and along with the mean values of each individual replicate (Fig 2D). Next, we used estimation statistics and bootstrapping [43–45] to compare test conditions to the control symmetric DMSO condition. This approach yields the 95% confidence intervals of the likely difference of the mean values of the worm position between a given test condition and the control (Fig 2E, mean difference). To understand the implications of pooling across replicates, we compared mean difference distributions derived by analyzing individual replicates (gray) and by pooling across them (color). We found that these 2 approaches generate average values that are indistinguishable from one another (Fig 2D and 2E) except that pooling narrows the confidence intervals. From these data, we also infer that DMSO is a weak attractant and confirm, as reported in many prior studies (reviewed in [4]), that isoamyl alcohol and diacetyl are strong attractants and that 2-nonanone and 1-octanol are strong repellents.

To evaluate the mean position as an indicator of chemosensitivity, we compared it to the chemotaxis index. Classically, researchers have reported the results of chemotaxis assays using a chemotaxis index: *chemotaxis index* $= (p − q) / (p + q)$ where $p$ is the number of animals on the side of the test chemical and $q$ is the number on the opposite or control side. Consistent with prior practice and to minimize the impact of variation in movement ability, animals in the starting zone were excluded from analysis (Methods). Comparing 3 biological replicates testing wild-type against 96 conditions consisting of 90 plant SMs, 2 null reference conditions, 2 attractants (isoamyl alcohol, diacetyl), and 2 repellents (2-nonanone, 1-octanol), we found that chemotaxis index and mean worm position were tightly correlated with one another (Fig 3, $R^2 = 0.966$), indicating that our analytical approach is consistent with classical studies. The tight correlation between these 2 measures is reinforced by prior work demonstrating that the aggregated response of many individual worms is similar to a group of worms [9]. Thus, the mean position is correlated with and essentially equivalent to the chemotaxis index.

We assessed OWL's performance by benchmarking the software against human scorers. To do this, we generated a test dataset and recruited 2 team members to manually tag the location of worms in each arena using FIJI [42]. The test dataset included 19 images of assays performed with 2 attractants (diacetyl, isoamyl alcohol), 2 repellents (2-nonanone, 1-octanol), symmetric solvent (DMSO:DMSO), and solvent (DMSO) opposite water (DMSO:water). To assess the agreement between the human observers and OWL, we compared the total number of worms (Fig 4A) (Pearson's correlation coefficient = 0.90) and mean worm positions (Fig 4B) (Pearson's correlation coefficient = 0.98). Whether measured by humans or OWL,

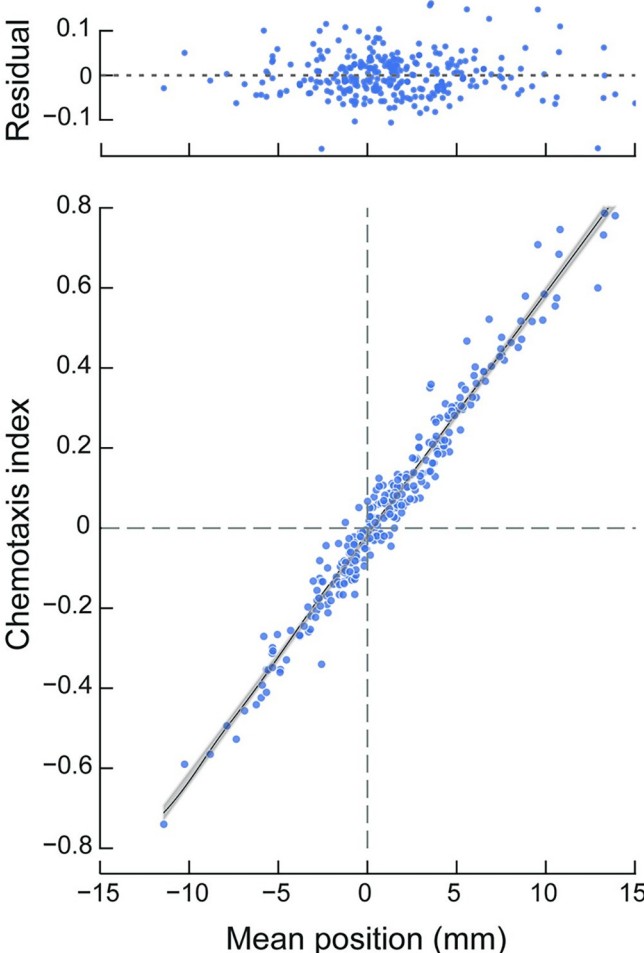

**Fig 3. Chemotaxis index and mean worm position are similar across a range of values and test conditions.** Each point represents the chemotaxis index and mean worm position computed from a single assay. The dataset represents 288 assays of the response of wild-type worms to 96 compounds ($N = 3$ biological replicates). Black line is a least-squares fit to the data with a slope of 0.06 ($R^2 = 0.97$), the gray shaded area shows the 95% confidence interval for the fit. The residuals of the fit (above) show the difference between the experimental and fitted values. Data used to calculate these statistics and generate this figure are reported in S2 Data.

strong attraction was more prevalent than strong repulsion (Fig 4B). The strong agreement between automated worm location and manual counting is similar to the findings of Crombie and colleagues [53] who paired large-particle sorting hardware (COPAS biosorter) with custom software to automate nematode chemotaxis assays performed on round Petri dishes. While OWL undercounted worms relative to human observers, human observers were also discordant (Fig 4A). Importantly, the average worm position measured by human observers was similar to that extracted by OWL. We suspect that the primary difference in worm counts resides in imperfect attempts by human observers to count aggregated animals. OWL excludes such aggregates (based on their size), a factor likely to account for the fact that humans find more worms. These effects are independent of position in the arena, however, since the distribution of worms as a function of position along the *y* axis is similar when measured by human observers and by OWL (Fig 4C). Thus, similar to the parallel worm tracker [52], the concordance between human observers resembles that found between a single human observer and OWL. In summary, the OWL image processing pipeline reliably determines

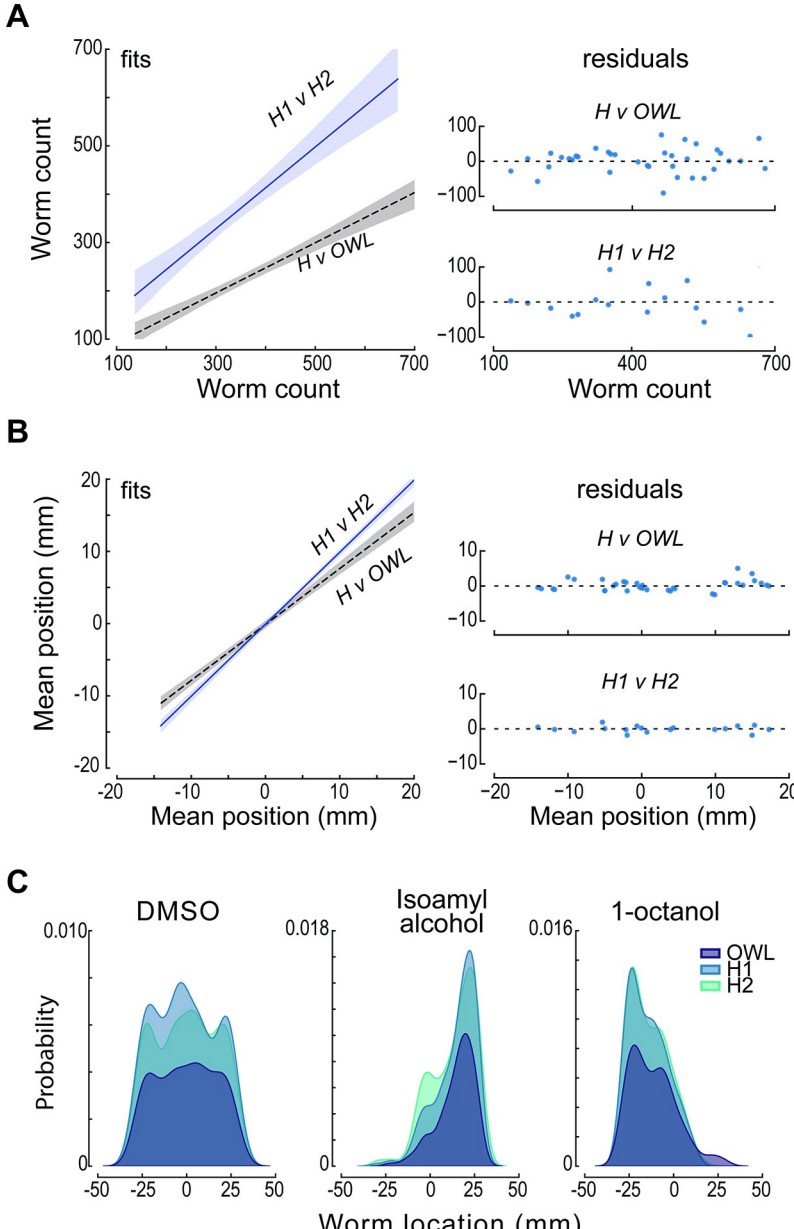

**Fig 4. Performance of human scorers and OWL software. (A)** Relationship (left) between the total number of worms detected by humans, H1 and H2 (solid blue line, slope = 0.85 $R^2$ = 0.83), and by the average human and OWL software (dashed black line, slope = 0.52; $R^2$ = 0.81). Shaded areas show the 95% confidence intervals of the fit. The fit residuals (right) indicate no systematic effect of the number of worms. **(B)** Relationship (left) between the mean worm position detected by H1 and H2 (solid blue line, slope = 0.99; $R^2$ = 0.99) and by the average human and OWL software (dashed black line, slope = 0.77; $R^2$ = 0.96). Shaded areas show the 95% confidence intervals of the fit. The fit residuals (right) indicate no systematic effect of the mean position. The test dataset shown in (**A**) and (**B**) was derived from images of 19 assays (4 of diacetyl and 3 for all other conditions). **(C)** Density as function of distance along the chemical gradient for 3 conditions (left to right): null condition (DMSO:DMSO), a known attractant (isoamyl alcohol), and a known repellent(1-octanol). Distributions scored by humans (light blue and aqua) and determined by OWL software (dark blue) are similar. Each image in the test dataset ($N$ = 3) was scored by 2 human experimenters and by the OWL software, as described in Methods. Data used to calculate these statistics and to generate this figure are reported in S3 Data.

average worm position, does not compromise reproducibility compared to pairs of human observers, and dramatically increases experimental throughput.

## Dozens of plant-derived small molecules attract or repel *C. elegans*

We applied our platform and integrated data handling workflow to screen 90 plant SMs and 6 reference conditions (isoamyl alcohol, diacetyl, 2-nonanone, 1-octanol, DMSO:DMSO, DMSO:water). A compound was considered chemoactive and worthy of additional study if the mean worm position observed in arenas containing that compound differed significantly from our 2 null reference conditions (DMSO:DMSO and DMSO:water). Using estimation statistics and bootstrapping [43,44], we computed the difference of the mean position for each compound relative to each of the null reference conditions. Fig 5 plots the distributions of mean differences (95% confidence intervals) and arranges the results by magnitude and valence such that the strongest attractants are at the top and the strongest repellents are at the bottom. Forty-one compounds in total (including 4 reference compounds) induced a response in which the 95% confidence interval of the mean difference relative to one or both null conditions excluded zero. In other words, each of these compounds produced a distribution that differed from one or both null conditions with $p < 0.05$. When accounting for multiple comparisons (Methods), 3 SMs that evoked responses were identified as potential false positives: oleanolic acid, sabinene, and sinomenine hydrochloride. Additionally, the library contained 3 pairs of SMs that were nominally identical (Fig 5, brown lines, text) obtained from different suppliers (Methods). For 2 of the 3 SM pairs, the response of wild-type worms to compounds were distinct from one another according to a Mann–Whitney U test: 2,3-dihydrobenzofuran and coumaran (CAS No. 496-16-2), $p = 8.3 \times 10^{-10}$; daucosterol and sitogluside (CAS No. 474-58-8), $p = 9.6 \times 10^{-4}$. These findings could reflect a true difference in the purity of the chemicals we tested. For the third pair, citronellol and β-citronellol (CAS. No. 106-22-9), the responses were indistinguishable ($p = 0.16$). Excluding references, 27 compounds attract wild-type worms and 10 repel them. Thus, our screening platform uncovers SMs that attract or repel wild-type *C. elegans* with high confidence and efficiency.

We next sought to determine which of these plant SMs had been tested previously for their ability to attract or repel *C. elegans* or other nematodes. To achieve this goal with similar coverage for all compounds, we used a defined keyword search of a standard bibliographic database (Methods). With the exception of 2 attractive compounds, furfural [21] and 2-methyl-1-butanol [55–57], we found that these plant SMs had not been tested for their activity in chemotaxis assays in *C. elegans* or any other nematode. We also searched for studies applying these SMs to *C. elegans* or other nematodes for any other purpose. Six compounds (phytol, ellagic acid, camphor, ursolic acid, furfural, and 2-methyl-1-butanol) have been tested for effects on life span, oxidative stress, fecundity, or as nematicides [58–62]. Three compounds, furfural, solasodine, and phytol, have been tested as tools for managing root-knot nematodes that parasitize plants, including important crops [63–68]. This raises the possibility that other compounds in this dataset may prove relevant to agriculture. More broadly, our systematic review buttresses the idea that combining an evolution-inspired screen design with an efficient phenotypic screening platform is a highly effective tool for discovering novel chemoactive natural products.

## Anosmic tax-4;osm-9 double mutants are indifferent to chemoactive SMs

To learn more about the genetic basis of chemotaxis valence, we tested these compounds against mutants lacking one or both of the 2 ion channel effectors required for chemosensory transduction (reviewed in [4]): TAX-4 and OSM-9. To do this, we relied on 2 previously isolated null mutants, *tax-4(p678)* [69] and *osm-9(ky10)* [16], and used them to generate an

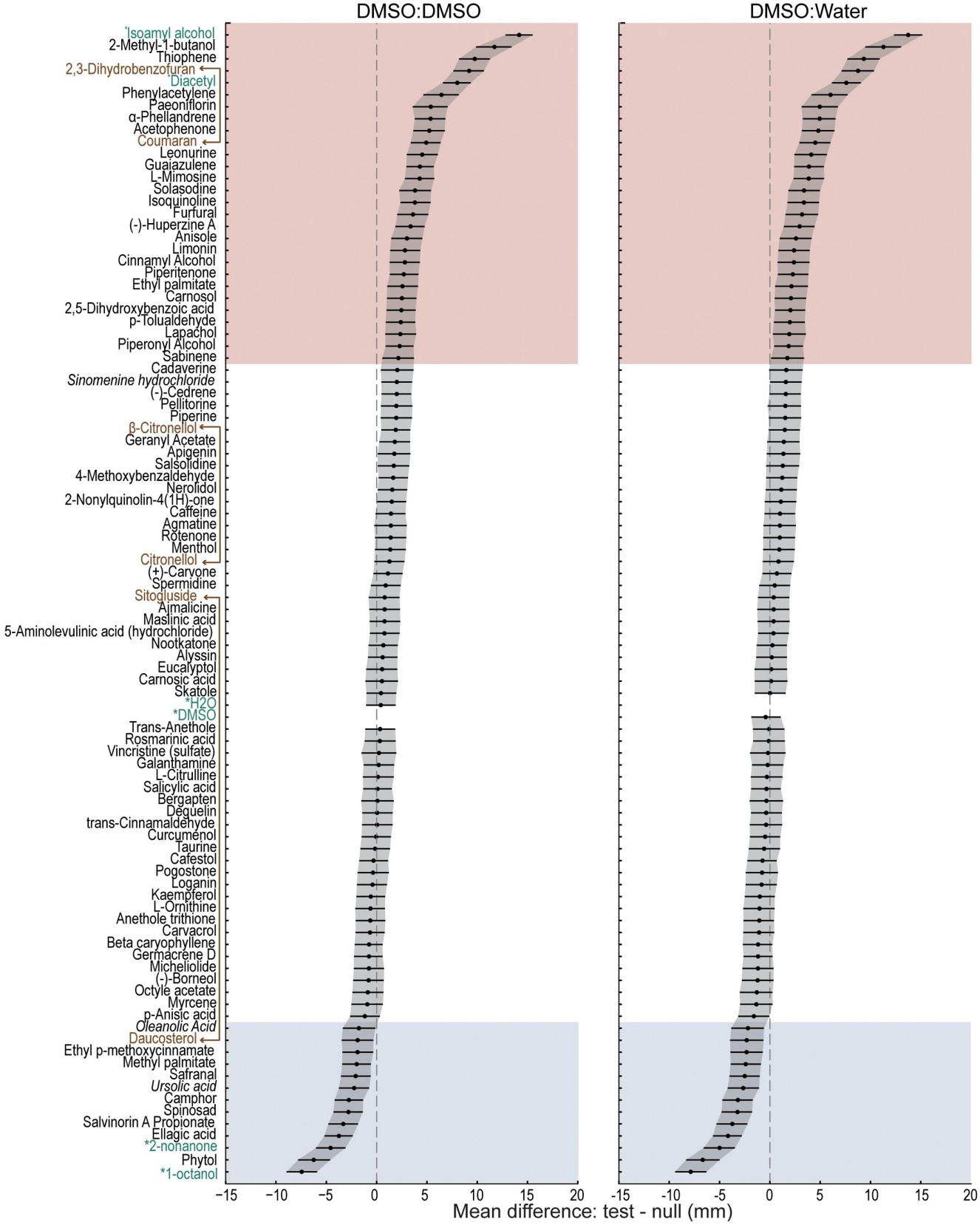

**Fig 5. A screen of 96 conditions reveals 37 SMs that are chemoactive in wild-type *C. elegans*, evoking either attraction (pink) or repulsion (blue).** The chemical panel contained 90 plant SMs and 6 reference conditions (green text, asterisks: isoamyl alcohol, diacetyl, 2-nonanone, 1-octanol, DMSO, and water). Results are sorted (top to bottom) according to the difference in mean position relative to 2 null reference conditions: symmetric DMSO:DMSO (left) and asymmetric DMSO:water (right). Positive values correspond to attraction and negative values correspond to repulsion. Black points and lines are, respectively, the difference of the mean position in each test condition relative to the reference condition and the 95% confidence intervals of these values. Shaded areas indicate putative attractants (pink) and repellents (blue). The panel includes 3 pairs of nominally identical compounds (brown text connected with solid lines) and 3 compounds (*italics*) eliciting weak responses likely to be false positives after correcting for multiple comparisons. S2 Table reports the sample size (*n* = worms pooled across *N* = 3 biological replicates), the difference of the mean position (in mm) for wild-type (N2) in experimental vs. reference conditions (DMSO: DMSO and DMSO:water), 95% confidence intervals (5% CI, 95% CI), exact *p*-values, and correction for multiple comparisons (5% FDR, B-H). Individual data points underpinning these measurements are reported in S4 Data.

anosmic *tax-4;osm-9* double mutant (Methods). Fig 6 shows responses of *osm-9;tax-4* (left), *osm-9* (center), and *tax-4* (right) mutants alongside those of wild-type animals (replotted from Fig 5). For all attractants and repellents, *tax-4;osm-9* double mutants were either indifferent or weakly repelled (Fig 6A, left). We used bootstrapping (Methods) to quantify this effect, color-coding the mean values for the difference between the response in wild-type and each mutant (ΔΔ). This analysis was repeated for all 3 mutant lines, and the results are overlaid on each panel. More saturated colors correspond to larger effects of each genotype on chemotaxis behavior and less saturated colors indicate that wild-type responses are similar to those found in the relevant mutant. This analysis yields 3 sets of ΔΔ (mutant—wild-type) values, which we used to position responses to SMs in a three-dimensional space (Fig 6B). The SMs are distributed within this space according primarily to response strength and valence (attraction and repulsion). Further classification awaits additional studies of the genetic basis of chemotaxis responses.

Chemotaxis to all SMs in our panel was altered in *tax-4;osm-9* anosmic mutants relative to wild-type animals (Fig 6A, left), with 3 exceptions: methyl palmitate and the triterpenoid isomers, ursolic acid, and oleanolic acid. These 3 SMs evoked weak repulsion in both wild-type animals and anosmic mutants, resulting in ΔΔ values close to zero (indicated by pale colors). Not all weak responses were similar in wild-type and anosmic mutants, however. For instance, the weak attraction seen following exposure to sabinene and simonene hydrochloride in the wild type was not evident in *tax-4;osm-9* double mutants, providing experimental evidence that, despite being flagged as putative false positive responses by statistical analysis (Fig 5 and S1 Table), these 2 compounds are genuine, if weak, attractants. These findings establish that the observed behaviors in response to most of the chemoactive compounds depend on known chemosensory signaling pathways and are unlikely to reflect indirect modulation of locomotion. Finally, they indicate that more than 30% of the compounds in our curated testing library of plant SMs are biologically active chemical cues in wild-type animals and imply that the *C. elegans* chemosensing repertoire is larger than previously appreciated.

## Loss of a single chemosensory ion channel subunit inverts chemotaxis valence

The chemosensory valence of 10 SMs was inverted in *osm-9* or *tax-4* mutants compared to the wild type (Fig 6, green ovals). Piperonyl alcohol attracts wild-type animals but repels *osm-9* mutants. Acetophenone strongly attracts wild-type animals and repels *tax-4* mutants, but *osm-9* single mutants and *tax-4;osm-9* double mutants were indifferent to this SM. Eight compounds were weak repellents of wild-type animals and weak attractants of *tax-4* mutants: oleanolic acid, daucosterol, methyl palmitate, ursolic acid, salvinorin A propionate, ellagic acid, spinosad, and phytol. These compounds evoked little or no response in *osm-9* single mutants and *tax-4;osm-9* double mutants (Fig 6, left and center). Due to their weak responses in wild-type animals, this group of compounds might have been overlooked, but for the observed valence inversion in *tax-4* single mutants. Finally, phytol is strongly repellent to wild-type worms and attractive to *tax-4* mutants. Phytol is an acyclic diterpene that is a

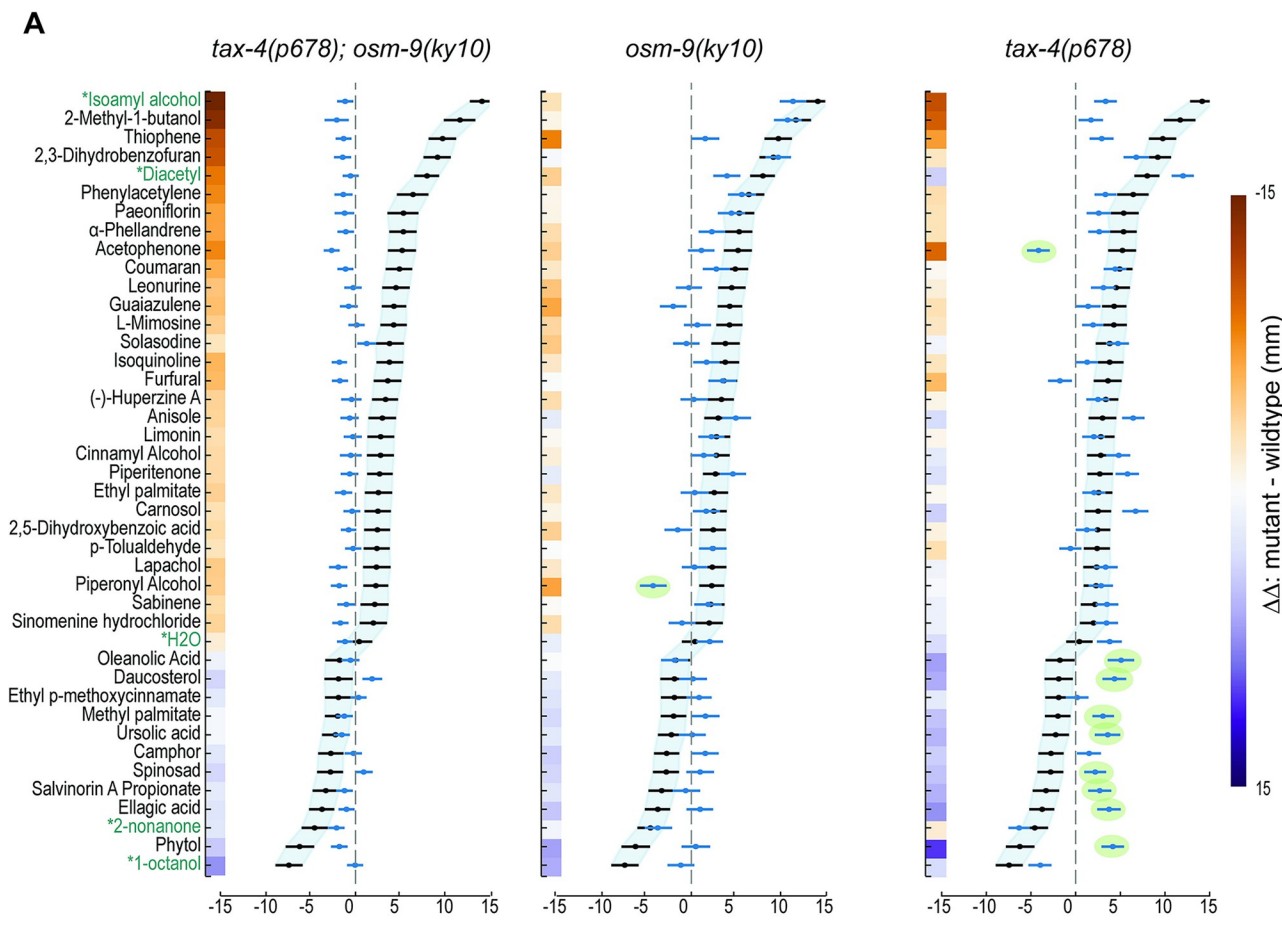

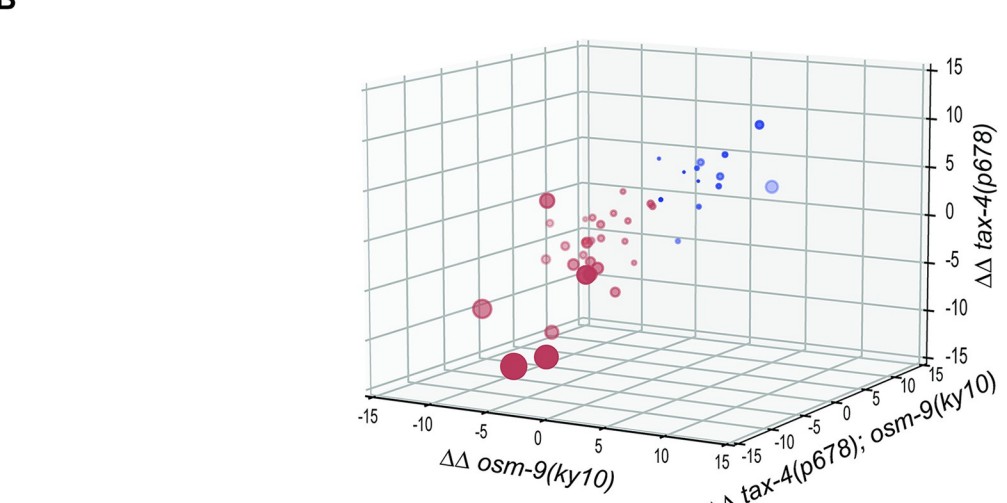

**Fig 6. Chemoactive plant SMs evoke approach or avoidance based on signaling by CNG channels, TRPV channels, or both chemosensory ion channels. (A)** Bootstrapped difference in the mean position (±95% confidence interval) for each plant SM tested in *tax-4(p678);osm-9(ky10)* (left), *osm-9(ky10)* (middle), and *tax-4(p678)* (right) mutants. Blue points and lines represent the difference in bootstrapped mean position (±95% confidence intervals) for SM responses in mutants relative to symmetric DMSO, while black points and lines shaded in light blue represent the wild-type (N2) values (reproduced from Fig 5). Green ovals encapsulate responses in single mutants that are opposite in sign (valence) compared to the wild type. We

computed ΔΔ values (mutant vs. wild type and SM vs. symmetric DMSO) via bootstrapping (Methods), encoded these values using the indicated color map, and displayed them along the *y* axis. (**B**) Three-dimensional plot of mean ΔΔ values for each mutant compared to wild type. SM valance is encoded in color: red symbols correspond to SMs that attract wild type, while blue symbols are SMs that repel wild type. The area of each symbol is proportional to the strength of attraction or repulsion; the more saturated the symbol color, the closer it is to the viewer in three-dimensional space. Thus, large, dark red symbols represent strong attractants with large negative ΔΔ values along the *tax-4;osm-9* and *osm-9* axes. Values plotted as points (mean difference) and lines (95% confidence intervals) in panel (**A**) are tabulated in S2 Table (wild type) and S3 Table (mutants: GN1077 *tax-4;osm-9*; CX10 *osm-9*; PR678 *tax-4*) along with sample size in worms pooled across 3 biological replicates. Mean ΔΔ values, 95% confidence intervals (5% CI, 95% CI) encoded in color bars in panel (**A**) and used to position SMs in the 3D space in panel (**B**) are reported in S4 Table. Data points underpinning these measurements are reported in S4 Data.

component of chlorophyll and is found in all photosynthetic organisms. From these findings, we infer that the wild-type chemosensory valence of these 10 SMs reflects integration of information from multiple signaling pathways. Since each of the 16 classes of CSNs expresses one or both TAX-4 and OSM-9 channel effectors (Fig 7A), integration could occur within single or across several CSNs and is likely to require multiple receptors for each ligand in this group of SMs.

## For most SMs, chemosensory signaling depends on both tax-4 and osm-9 ion channel genes

The ability to measure responses to a large panel of chemoactive SMs against 4 genotypes provides an opportunity to determine what response patterns occur most frequently. To reach this goal, we computed ΔΔ values (using bootstrapping; Methods) comparing responses in pairs of genotypes, including those shown in Fig 6 (*tax-4;osm-9* versus wild type; *tax-4* versus wild type; *osm-9* versus wild type) and extended this approach to compare responses seen in each of the single mutants against *tax-4;osm-9* double mutants (S4 Table). Across SMs, we quantified the effect of each mutant relative to chemosensitive wild-type animals or to anosmic *tax-4;osm-9* double mutants in a valence-agnostic manner using the absolute value of the computed ΔΔ values. We binned the entire range of |ΔΔ| values (min, max = 0.02, 15.32 mm) into quartiles and used these values to classify response patterns. SMs that generated similar behaviors in the genotypes under comparison had |ΔΔ| values less than the median (3.02 mm). And, SMs generating substantially distinct responses in the genotypes under comparison had values larger than the median. In this framework, SM responses that primarily depend on *tax-4* signaling induce the following: (1) substantial (>median) differences between *tax-4* and wild type; (2) modest (<median) differences between *tax-4* and anosmic mutants; and (3) modest (<median) effects of *osm-9* relative to wild-type response. The logical equivalent for *osm-9*-dependent signaling is that the SM induces (1) substantial differences *osm-9* and wild-type responses, (2) modest differences in responses in *osm-9* and the anosmic mutants, as well as (3) modest differences between responses in *tax-4* mutants and wild-type animals.

Based on this rubric, we classified these SMs as reliant primarily on a single chemosensory ion channel (*tax-4* or *osm-9*) (Fig 7Bi and 7Bii) or reliant on both chemosensory ion channels (Fig 7Ci and 7Cii). Effects sizes are encoded as a continuous color map covering the entire range of |ΔΔ| and tabulated in S4 Table. Responses to only 8 chemoactive SMs satisfied the criteria for being primarily reliant on a single chemosensory ion channel (Fig 7B). Only a single SM evoked responses qualified as *tax-4* dependent and *osm-9* independent: furfural (Fig 7Bi). This result reinforces prior work showing that furfural functions as a chemoattractant [21] and suggests that chemotaxis responses that depend primarily on *tax-4* are uncommon. Many more SMs qualified as primarily *osm-9* dependent and largely *tax-4* independent: solasodine, 2,5-dihydroxybenzoic acid, L-mimosine, leonurine, guaiazulene, 1-octanol, and thiophene (Fig 7Bii).

Responses to the remaining chemoactive SMs displayed a variety of response patterns (Fig 7Ci). For instance, avoidance of camphor required both *tax-4* and *osm-9* genes since loss of

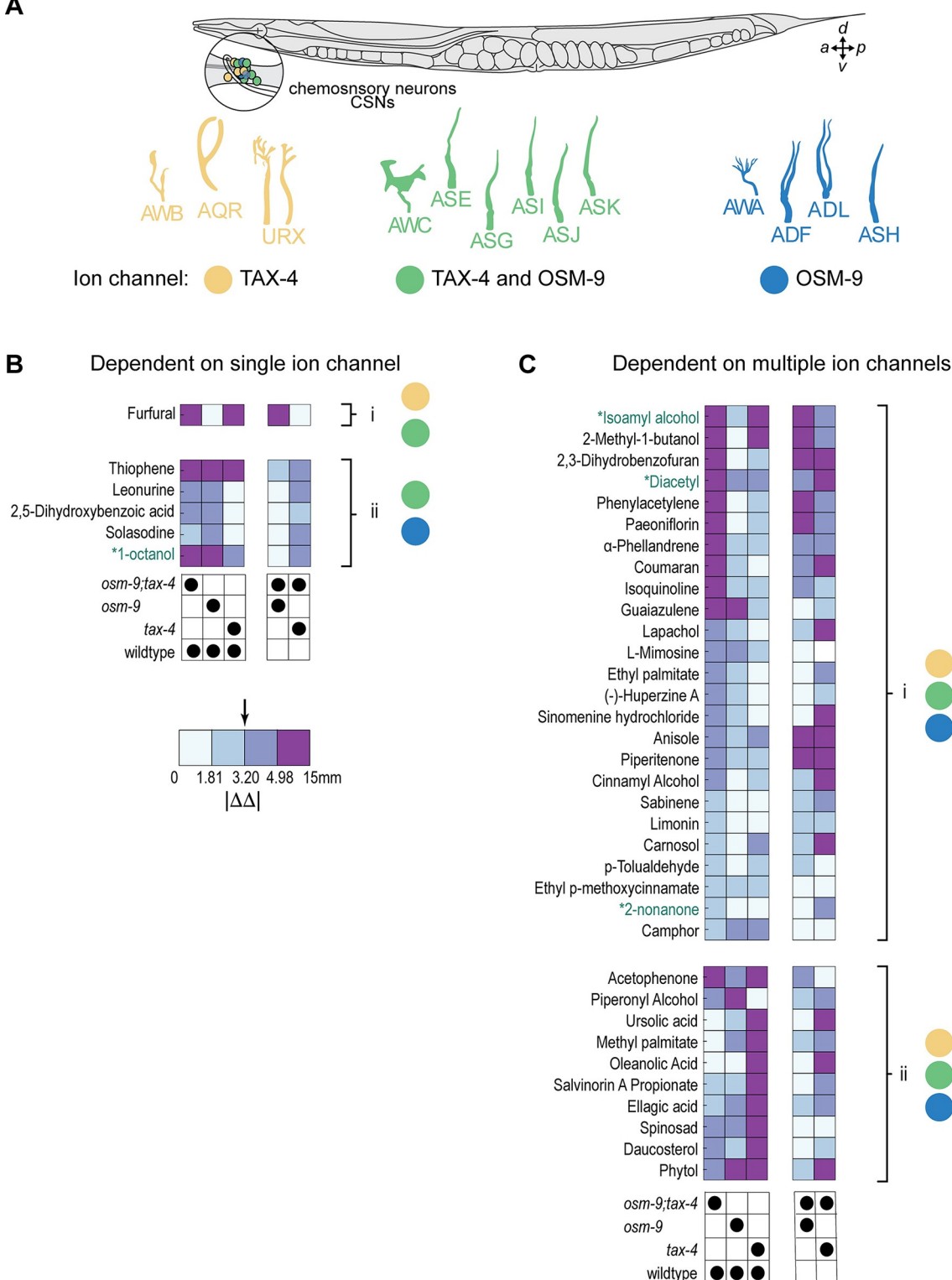

**Fig 7. Graphical summary of behavioral responses to chemoactive compounds and proposed links to candidate chemosensory neurons (CSNs). (A)** Schematic showing the position of *C. elegans* anterior CSNs on the right side of an adult animal (top). With the exception of AQR, CSNs are bilaterally symmetric. CSNs have distinctive cilia, shown schematically (bottom). Illustration adapted from [70]. Color indicates expression of chemosensory transduction ion channels in each CSN, where yellow, blue, and green highlight CSNs expressing *tax-4*, *osm-9*, or both ion channel genes, respectively. **(B)** SM responses primarily dependent on *tax-4* (i)

and *osm-9* (ii) based on how responses are modified by mutations. Each column in the heatmap represents the |ΔΔ| values for the pairs of genotypes indicated below. (**C**) SM responses dependent on both *tax-4* and *osm-9* (i) or that invert valence in single mutants (ii). The color bar delineates the range of effect sizes binned into quartiles and numbers indicate values separating quartiles. The arrow denotes the median of the effect sizes where values to the right (>median) have a larger effect and values to the left (<median) indicate little to no effect. |ΔΔ| values used to determine the quartiles in panel (**B**) and panel (**C**) are tabulated in S4 Table. Individual data points underpinning the measurements plotted in this figure and tabulated in S4 Table are reported in S4 Data.

either channel produced responses similar to those found in the anosmic mutant lacking both channels. In other cases, such as attraction to limonin, the 2 ion channel effectors appeared to be redundant: loss of either channel resulted in responses indistinguishable from wild type, but knocking out both abolished the observed response. In other cases, loss of either ion channel decreased, but did not abolish, the behavioral response (for instance, α-phellandrene).

This group of SMs also includes 3 reference compounds (diacetyl, isoamyl alcohol, and 2-nonanone) that evoked strong responses that were reduced in *tax-4* and *osm-9* single mutants relative to wild-type and *tax-4;osm-9* double mutants. Consistent with this finding, the attractants diacetyl and isoamyl alcohol evoke calcium transients in neurons that express both *tax-4* and *osm-9* [71,72]. The repellent 2-nonanone evokes calcium transients in the *osm-9* expressing ASH neuron and in the *tax-4* expressing AWB neuron [73]. This study of chemotaxis and complementary calcium imaging [71,72] suggest that the animal's ability to classify SMs as desirable or potentially toxic emerges from the actions of multiple CSNs.

Based upon the pattern of phenotypes evident in the 4 genotypes we tested and the cellular expression patterns of the *tax-4* and *osm-9* ion channel genes, we draw inferences regarding the CSNs likely to detect the chemoactive compounds. As illustrated in Fig 7A, the *tax-4* and *osm-9* genes are coexpressed in 6 anterior CSNs: AWC, ASE, ASG, ASI, ASJ, and ASK. The AWB, URX, and AQR CSNs express *tax-4* but do not appear to express *osm-9*. And, the AWA, ADF, ASH, and ADL neurons express *osm-9* but do not appear to express *tax-4* [15,16]. Considering only compounds that generated responses in single ion channel mutants that are distinct from the wild type and from *tax-4;osm-9* doubles, we infer that 6 compounds (furfural, thiophene, leonurine, 2,5-Dihydroxybenzoic acid, solasodine, and 1-octanol) are detected by at least 1 CSN using either TAX-4 or OSM-9 as the primary effector. Further, we propose that 35 compounds are detected by at least 2 CSNs using TAX-4, OSM-9, or both ion channels as effectors. Although additional experimental work is needed to link individual plant SMs to CSNs and to their membrane protein receptors, the ability to screen a large panel of SMs against 4 genotypes demonstrated here reveals that *C. elegans* chemotaxis is more likely to depend on integration of information contributed by multiple CSNs and ligand–receptor pairs than it is to arise from signals delivered by a single class of CSN.

## Discussion

To expand knowledge of the nematode chemical-sensing repertoire and to spur efforts toward obtaining a general understanding of how chemical cues are encoded according to valence, we built an efficient platform for testing the ability of SMs to attract or repel nematodes. Compared to classical *C. elegans* chemotaxis assays, which depend on manual assays and worm counts [4,22], our platform features liquid handling hardware for worm dispensing, flatbed scanners for rapid image acquisition, and modifications to optimize image quality and enable image demultiplexing. Software to count animals, determine their position, and determine the strength and direction of chemotaxis and integrated data management completes the system. The workflow presented here makes it possible to screen hundreds of compounds in a single week with improved rigor and reproducibility. Across >250 assays, we demonstrate that mean worm position is equivalent to the classical chemotaxis index (Fig 3C). Recording worm position in a

standardized, open-source digital data format opens the door to pooling results across replicates. This tactic also generates improved statistical power and is amenable to using estimation statistics to determine the effect size relative to reference compounds and null conditions [43,44].

Our chemotaxis assay platform and integrated OWL software are versatile and compatible with any desired chemical library. Based on the long cohabitation of nematodes and plants, we reasoned that screening a library of plant-synthesized SMs would be especially productive and we screened a modest custom library of 90 plant SMs. Consistent with this evolution-inspired concept, we found that, relative to solvent controls, 37 of 90 or 41% of our curated plant SM library evoked chemotaxis in wild-type *C. elegans*. This group included 27 attractants and 10 repellents (Fig 5), 8 of which produced visible precipitates on assay arenas (S1 Table). Since the parent library contained a similar proportion of SM precipitates (17 of 96), compounds with this property were neither depleted nor enriched among chemoactive SMs. The overall preponderance of attractants could reflect an unintended bias in our library, masking of repulsion by the weak attraction induced by DMSO, or a true reflection of the bias in chemical communication between plants and nematodes. Regardless of its origin, a similar bias in favor of attractants was noted previously [21]. These responses require expression of the TAX-4 or OSM-9 (or both) chemosensory ion channels (Fig 6). Finally, most of the SMs identified as being chemoactive in this study had not been tested previously in *C. elegans* chemotaxis assays. Thus, the chemoactive SMs identified here expand the set of chemical cues known to evoke either positive or negative chemotaxis based on sensing by one or more *C. elegans* CSNs.

## Valence depends on the integration of multiple signaling pathways

How does response valence emerge? For many chemical cues studied here and elsewhere, chemotaxis behavior engages overlapping sets of CSNs and depends on dual chemosensory transduction pathways. To learn more about how worms encode the valence of chemical cues, we analyzed responses in single mutants lacking either TAX-4 or OSM-9, the ion channels responsible for chemosensory transduction (reviewed in [4]). Responses to more than half of the tested SMs were disrupted in both single mutants, indicating that behavioral valence most often reflects the integration of multiple chemosensory transduction pathways. Consistent with this inference, well-characterized attractants and repellents modulate calcium signaling in multiple CSNs [71]. For instance, the classical attractants isoamyl alcohol and diacetyl activate ASG and ASK, respectively, and both chemicals activate AWA, AWC, ASE, and ASH [71,72]. Here, we show that loss of *tax-4* impairs attraction to isoamyl alcohol and enhances attraction to diacetyl (Fig 6). Conversely, loss of *osm-9* has little impact on attraction to isoamyl alcohol and reduces attraction to diacetyl (Fig 6). From these findings, we infer that these 2 attractants are detected by distinct molecular signaling pathways. Despite their shared valence, the presence of these chemicals is transformed into action based on signals generated by distinct, but overlapping sets of CSNs. Notably, these sets of neurons are not uniquely activated by attractants. Indeed, all of the CSNs activated by isoamyl alcohol and diacetyl are also activated by the classical repellent, 1-octanol [71]. Avoidance of 1-octanol depends primarily on *osm-9*-dependent signaling (Fig 6), even though *osm-9* expression is evident in only some of the 1-octanol-sensitive CSNs. Notably, response valence was inverted in single *tax-4* or *osm-9* mutants compared to wild type in more than one-fourth (10 of 37) of the tested SMs. For instance, we found that phytol repels wild-type *C. elegans* but attracts *tax-4* single mutants. Phytol has no detectable effect on either *osm-9* mutants or *tax-4;osm-9* double mutants. We observed an analogous response pattern for acetophenone, which attracts wild-type *C. elegans*, repels *tax-4* single mutants, and has little or no effect on *osm-9* single mutants and *tax-4;osm-9* double mutants. In other words, wild-type avoidance of phytol (or attraction to acetophenone)

depends on an *osm-9*-dependent avoidance (or attraction) signal that supersedes a *tax-4*-dependent attraction (avoidance) signal. The scope of our screen reveals that complex encoding of behavioral valence is not rare, results that are aligned with calcium imaging studies of the responses to chemical cues [71] and suggest that studies examining panels of chemical cues will be needed to fully decipher how behavioral valence is encoded.

### Some plant SMs detected by *C. elegans* are chemical cues for other animals

Several of the chemoactive SMs we identified are synthesized by additional organisms or known to affect other nematode species. For instance, 2-methyl-1-butanol is produced by bacteria, yeast, and a variety of plants [74]. It is also used by the nematode-eating fungus, *Arthobotrys oligospora*, to attract nematodes [55] and as a sex pheromone in longhorn beetles [57]. Thus, this simple compound is a multifunctional chemical cue in nature and likely functions as a ligand for receptors present in multiple phyla. Whether or not the receptors themselves are conserved is an open question. Spinosad, a mixture of 2 complex macrocyclic lactones, is also produced by bacteria and is approved for use as an insecticide in purified form [75]. Our findings indicate that *C. elegans* is attracted to spinosad, although whether or not it is toxic to nematodes remains to be determined. Nevertheless, our findings suggest that the use of spinosad as an insecticide may have unintended consequences for nematode communities. Furfural, which attracts wild-type *C. elegans*, has been tested as a tool for managing *Meloidogyne incognita* [63,64], a root knot parasitic nematode that is a serious threat to agriculture. Phytol and methyl palmitate are other SMs in our collection that repel both *C. elegans* and root knot nematodes [68,76]. Camphor repels *C. elegans* (Fig 5) but attracts root knot nematodes [77]. Thus, sensitivity to some plant SMs is conserved among nematodes and might be exploited by their predators or mutualists in nature. These findings also highlight the potential using *C. elegans* as a tool to screen for natural products that may aid in managing parasitic nematodes.

### Several plant SMs detected by *C. elegans* are ligands for human GPCRs or ion channels

Numerous precedents suggest that plant SMs include ligands for GPCRs in *C. elegans* and humans. For instance, morphine, which is synthesized by the opium poppy, activates GPCRs in humans [78] and in *C. elegans* [79]. Consistent with this precedent, 8 plant SMs that evoke *C. elegans* chemotaxis in wild-type animals, but not *tax-4;osm-9* double mutants, are also listed as ligands for at least 1 human or mouse GPCR in online databases [80,81]: acetophenone, anisole, camphor, cinnamyl alcohol, ellagic acid, methyl palmitate, oleanolic acid, and ursolic acid. Acetophenone activates 11 human olfactory GPCRs and 78 mouse olfactory GPCRs [82]. These GPCRs share a set of residues predicted to form the orthosteric binding pocket for acetophenone, but the proteins themselves are not otherwise considered orthologs or paralogs [82]. Our finding that acetophenone attracts wild-type *C. elegans* and repels *tax-4* mutants (Figs 6 and 7) implies that there are also at least 2 acetophenone receptors in *C. elegans*. The weak repellents, ellagic acid and methyl palmitate, activate human GPR35 and the CB1/2 receptors [83,84], respectively, and oleanolic acid and ursolic acid both activate GPBAR1 (aka TGR5) [85,86]. Thus, the ability to detect and respond to individual plant SMs is conserved among animals as distantly related as nematodes, rodents, and humans. It is tempting to speculate that, regardless of the animals producing GPCRs, a shared ability to detect a given SM reflects the presence of receptors bearing structurally similar ligand binding pockets.

It remains to be determined if plant SM-evoked nematode attraction and repulsion is mediated primarily or exclusively by GPCRs, although at least 1 well-characterized attractant, diacetyl, has been linked to 2 GPCR genes [32,87] and responses to several other chemical cues

require one or more G proteins expressed in CSNs [88]. However, several plant SMs that evoke attraction or repulsion are known to modulate ion channels in other animals. Huperazine A, which is a *C. elegans* attractant (Fig 5), modulates ionotropic acetylcholine and glutamate receptors [89]. Camphor, a weak repellant, is a well-characterized agonist for TRPV3 channels [90], and limonin, a weak attractant, blocks the human TMEM16A calcium-activated chloride channel [91]. Thus, more than one-quarter of the plant SMs identified here as either *C. elegans* attractants or repellents also bind to one or more membrane proteins in other animals, including mammals. These compounds comprise more than 10% of the library that we screened, and these findings suggest that further screening is likely to yield additional ligands for membrane proteins in *C. elegans* and humans.

## Limitations and future research

Chemical cues are widespread in nature and used by most, if not all animals to locate food and avoid harm. Our platform is simple, delivering all test compounds at a single concentration. This design choice limits the inferences that we might draw regarding response strength and might result in a failure to detect some bona fide responses. It might also affect response valence, since some chemical cues are attractive to wild-type *C. elegans* at low concentrations and repulsive at higher ones [31–33]. On a similar note, we captured responses at a single time point (1 hour) and worms might habituate during this time, affecting the measured strength or valence of the response. Previous studies have shown that over the span of 1 hour, valence changes over time for 1 compound, benzaldehyde, but not for another, diacetyl [34]. Thus, it is possible that our screen omits some chemical cues or inverts responses to others. Future studies could provide insights into these questions by testing compounds across a range of concentrations or assay durations.

Like all chemotaxis assays, the platform presented here is affected by variations in compound stability and their interaction with solid media. Some SMs may be sensitive to light exposure, humidity, and temperature, while others may be present in a mixture of protonated and deprotonated forms based on their $pK_a$ relative to the pH of the buffer incorporated into the solid media, and still others may be particularly hygroscopic or hydrophobic. These physicochemical factors as well as variations in diffusion constants could reduce the effective SM concentration or alter the nature of the chemical gradient established in each assay arena. Because we did not explicitly examine the impact of these factors in this work, it is therefore possible that a subset of SMs that did not appear to affect *C. elegans* behavior in this study might evoke attraction or repulsion under different conditions.

The platform design is compatible with any chemical library and with other nematode species. Applicable nematodes include both lab-reared and wild *C. elegans* strains and other species that can be maintained in the laboratory, including parasites of plants and animals. Thus, this platform could be adapted to support discovery of chemical tools for control of parasitic nematodes or chemical actuators of the nervous system. Indeed, 6 of the chemoactive compounds studied here are annotated as relevant to neurological disease [92]: carnosol, huperizine A, leonurine, l-mimosine, acetophenone, and paeoniflorin. Whereas this study and many others primarily evaluate responses to pure compounds, natural chemical cues are present in complex mixtures. Fortunately, this experimental workflow can readily extend to experimenter-defined mixtures, extracts of natural products obtained from plants, fungi, and bacteria, or even to colonies of microorganisms. With advanced liquid handling, it would become practical to determine the chemical valence exhibited by several nematode species or a collection of *C. elegans* strains in parallel. For example, these tools would enable the simultaneous evaluation of responses of divergent nematode strains to a common chemical library and make it possible to evaluate the covariance of chemotaxis and genetic variation. Combining

this approach with advances in high-throughput tracking of freely moving animals and imaging CSN responses would deepen our understanding of the mechanisms underpinning the emergent property of chemotaxis valence.

## Supporting information

**S1 Fig. Workflow infographic of *C.elegans* chemotaxis screening platform.** Timeline (in minutes) shown from top to bottom, time points (circles) and actions are indicated to the left and right of the timeline. Created with BioRender.com.
(PDF)

**S2 Fig. Schematic showing integrated data management and image analysis.** Data management (left) and image analysis (right) for the screens occur simultaneously, reducing data processing time, reducing data processing errors, and increasing reproducibility. Created with BioRender.com.
(PDF)

**S1 Table. Screening library.** List of small molecules comprising the curated screening library, including the CAS registry number (aka CAS No.), common name used in this study, vendor, and catalog number. Vendors are (alphabetical order): Ambeed, Arlington Heights, IL; Aobious, Gloucester, MA; Cayman Chemical, Ann Arbor, MI; Chem-Impex, Woodale, IL; MCE = MedChemExpress, Monmouth Junction, NJ; Sigma-Aldrich, St. Louis, MO; Target-Mol, Boston, MA; TCI = TCI America, Portland, OR; VWR International, Radnor, PA. Compounds generating visible precipitates in assay arenas are indicated with "(p)".
(PDF)

**S2 Table. Responses of wild-type *C. elegans* to compounds listed in S1 Table.** Tabulated list of the difference of the mean position for wild type between each test condition and a reference condition (aka "mean difference"), sample size ($n$ = worms pooled across $N$ = 3 biological replicates), 95% confidence intervals for the mean difference (5% CI, 95% CI), and statistical testing (exact $p$-values, B-H correction for multiple comparisons with a false discovery rate of 5%). Control sample size was $n$ = 1,065 for DMSO:DMSO, and $n$ = 915 for DMSO:water for the respective comparisons. Mean differences and confidence intervals obtained by bootstrapping using the Dabest statistical package [43]. These data are shown graphically in Fig 5 and are from assays conducted with wild-type (N2, Bristol) adult worms.
(PDF)

**S3 Table. Responses of mutant *C. elegans* worms to compounds listed in S1 Table.** Tabulated list of the difference of the mean position for mutant worms between each test condition and a reference condition (aka "mean difference"), sample size ($n$ = worms pooled across $N$ = 3 biological replicates), 95% confidence intervals for the mean difference (5% CI, 95% CI). Control (DMSO) sample size was $n$ = 851 for *tax-4(p678)*, $n$ = 936 for *osm-9(ky10)*, and $n$ = 915 *tax-4(p678)*; *osm-9(ky10)* for all comparisons. Mean differences and confidence intervals obtained by bootstrapping using the Dabest statistical package [43]. These data are shown graphically in Fig 6 and are from assays conducted with *tax-4(p678)*, *osm-9(ky10)*, and *tax-4 (p678)*; *osm-9(ky10)* adult worms.
(PDF)

**S4 Table. Responses to chemoactive compounds as a function of genotype.** Tabulated list of the |ΔΔ| values for each test condition and pairwise comparisons of the indicated strains (Strain1, Strain2), and 95% confidence intervals for the |ΔΔ|. The |ΔΔ| and confidence intervals were obtained by bootstrapping *via* the Dabest statistical package [43]. Strain [genotype]: N2

[wild-type], GN1077 [*tax-4(pr678);osm-9(ky10)*], CX10 [*osm-9(ky10)*], and PR678 [*tax-4 (p678)*].
(PDF)

**S1 Data. Data (worm positions) used to perform bootstrapping and generate summary statistics plotted in Fig 2.**
(XLSX)

**S2 Data. Data (worm position) and summary statistics used to generate Fig 3.**
(XLSX)

**S3 Data. Data (worm position) and summary statistics used to generate Fig 4.**
(XLSX)

**S4 Data. Data (worm position) to generate bootstrapped mean differences and bootstrapped delta-delta values in Figs 5–7.**
(XLSX)

## Acknowledgments

We thank J. Casar, A. Das, and L. O'Connell for contributing to the prototyping team; C. Jaisinghani for assistance with genetics; S. R. Lockery for suggesting foam sheets to define behavioral arenas; J. A. Franco for assistance with data management and visualization; and Z. Liao for research support and safety management. We also thank the Caenorhabditis Genetics Center (CGC), which is funded by NIH Office of Research Infrastructure Programs (P40 OD010440), for *C. elegans* strains.

## Author Contributions

**Conceptualization:** Emily Fryer, Sujay Guha, Lucero E. Rogel-Hernandez, Theresa Logan-Garbisch, Shaul Druckmann, Thomas R. Clandinin, Seung Y. Rhee, Miriam B. Goodman.

**Data curation:** Emily Fryer, Sujay Guha, Theresa Logan-Garbisch, Ehsan Rezaei, Adam L. Nekimken, Angela Xu.

**Formal analysis:** Emily Fryer, Sujay Guha.

**Funding acquisition:** Sujay Guha, Lucero E. Rogel-Hernandez, Thomas R. Clandinin, Seung Y. Rhee, Miriam B. Goodman.

**Investigation:** Emily Fryer, Sujay Guha, Lucero E. Rogel-Hernandez, Theresa Logan-Garbisch, Hodan Farah, Iris N. Mollhoff, Sylvia Fechner, Miriam B. Goodman.

**Methodology:** Emily Fryer, Sujay Guha, Lucero E. Rogel-Hernandez, Theresa Logan-Garbisch, Ehsan Rezaei, Adam L. Nekimken, Sylvia Fechner, Thomas R. Clandinin, Seung Y. Rhee.

**Project administration:** Emily Fryer, Sujay Guha, Lucero E. Rogel-Hernandez, Theresa Logan-Garbisch, Sylvia Fechner.

**Resources:** Emily Fryer, Sujay Guha, Lucero E. Rogel-Hernandez, Hodan Farah, Ehsan Rezaei, Sylvia Fechner.

**Software:** Emily Fryer, Ehsan Rezaei, Adam L. Nekimken, Lara Selin Seyahi.

**Supervision:** Thomas R. Clandinin, Seung Y. Rhee, Miriam B. Goodman.

**Validation:** Emily Fryer, Sujay Guha, Lucero E. Rogel-Hernandez, Theresa Logan-Garbisch, Thomas R. Clandinin, Seung Y. Rhee, Miriam B. Goodman.

**Visualization:** Emily Fryer, Sujay Guha, Lucero E. Rogel-Hernandez, Theresa Logan-Garbisch, Lara Selin Seyahi.

**Writing – original draft:** Emily Fryer, Sujay Guha, Lucero E. Rogel-Hernandez, Theresa Logan-Garbisch, Ehsan Rezaei, Thomas R. Clandinin, Seung Y. Rhee, Miriam B. Goodman.

**Writing – review & editing:** Emily Fryer, Sujay Guha, Lucero E. Rogel-Hernandez, Theresa Logan-Garbisch, Ehsan Rezaei, Iris N. Mollhoff, Adam L. Nekimken, Thomas R. Clandinin, Seung Y. Rhee, Miriam B. Goodman.

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
