## [Editor Report · Decision Letter 0]

11 Sep 2023

Dear Dr Goodman, 

Thank you for your Appeal regarding your manuscript entitled "An efficient behavioral screening platform classifies natural products and other chemical cues according to their chemosensory valence in C. elegans" for consideration as a Research Article by PLOS Biology.

Your Appeal has now been evaluated by the PLOS Biology editorial staff, as well as by an academic editor with relevant expertise, and I am writing to let you know that we would like to grant your Appeal and send your submission out for external peer review.

IMPORTANT: Please change your article type to "Methods and Resources" when you upload your additional metadata (see next paragraph).

Once your full submission is complete, your paper will undergo a series of checks in preparation for peer review. After your manuscript has passed the checks it will be sent out for review. To provide the metadata for your submission, please Login to Editorial Manager (https://www.editorialmanager.com/pbiology) within two working days, i.e. by Sep 13 2023 11:59PM.

Kind regards,

Roli Roberts

Roland G Roberts PhD

Senior Editor

PLOS Biology

rroberts@plos.org

on behalf of

Richard Hodge, 

Senior Editor

PLOS Biology

rhodge@plos.org

---

## [Decision Letter · Decision Letter 1]

1 Nov 2023

Dear Dr Goodman,

Thank you for your patience while your manuscript "An efficient behavioral screening platform classifies natural products and other chemical cues according to their chemosensory valence in C. elegans" was peer-reviewed at PLOS Biology. Please accept my apologies for the delays that you have experienced during the peer review process. Your manuscript has now been evaluated by the PLOS Biology editors, an Academic Editor with relevant expertise, and by three independent reviewers. 

In light of the reviews, which you will find at the end of this email, we would like to invite you to revise the work to thoroughly address the reviewers' reports.

As you will see, the reviewers are positive about your method to measure chemotaxis in C. elegans and think that the manuscript is comprehensive and well-done. However, the reviewers note that some important control measurements are using to fully validate the method, such as controlling for the effects of temperature due to the plate scanning and correcting for multiple comparisons in statistical tests to test for variance. In addition, Reviewer #2 notes that additional methodological reporting details and discussions focusing on the limitations should be included. 

Given the extent of revision needed, we cannot make a decision about publication until we have seen the revised manuscript and your response to the reviewers' comments. Your revised manuscript is likely to be sent for further evaluation by all or a subset of the reviewers.

**IMPORTANT - SUBMITTING YOUR REVISION**

*Re-submission Checklist*

*Published Peer Review*

*PLOS Data Policy*

*Blot and Gel Data Policy*

Sincerely,

Richard

Richard Hodge, PhD

rhodge@plos.org

REVIEWS:

Reviewer #1: Fryer et al. present a powerful new quantitative platform for measuring chemotaxis in C. elegans. This platform was optimized using a series of experimental and computational techniques that also combine creative custom-fabricated solutions. This approach was impressive and presents a method (and experimental paradigm) that other labs can mimic and follow. The use of open-source tools and python were laudable. Beyond the technique, the authors test 90 different plant-derived small molecules for chemotaxis and find that ~40% elicit effects on the C. elegans N2 strain. Their results also have implications for how chemosensory neurons recognize compounds and elicit organismal behaviors (attraction or repulsion). I have only minor points and some specific edits to address. The manuscript is fun and impressive.

Minor points:

(1) C. elegans is not a soil nematode. It is found in rotting fruits, flowers, tubers, etc. A variety of citations from the Felix and Andersen labs have shown this focus, so the two citations, which do show global distribution, should be amended or others added. Also, the section on page 11 dealing with the soil matrix should be edited. It is not clear from any study that they are found in the soil any more than the substrate on top of the soil. 

(2) In the Methods, the authors state "independent biological replicates", but these words can mean many different things to different researchers. Please add specificity about the growth, bleach, synchronization, etc. independence of each sample. Additionally, pooling across all three replicates will conflate effects of the preparation technique. Please perform an ANOVA to measure the variance from the different preparation parameters in the experimental set up.

(3) Like most chemotaxis index calculations, animals near the starting zone are excluded from the analysis. How big of a region was excluded around the band of the starting position?

Specific edits:

(1) Page 11, "evolution-inspired approach" is not obvious to me. As a process or approach, use of the hypothesized natural interactions is not evolution. Perhaps natural environment is more appropriate.

(2) Page 19, the equation is not rendered correctly in the Methods PDF and could not be evaluated.

(3) Throughout, the figure insertion locations were left accidently as notes in the manuscript text. 

(4) Page 31, "were disrupted in both single of the mutants" should be "in both single mutants"

(5) Page 31, "This findings indicate" should be "These"

Reviewer #2: Fryer et al present a new accessible high throughput method for screening the valence of C. elegans chemotaxis to large libraries of small molecules. The approach presents advances in terms of the integration of liquid handling, plate-reading and computer-vision and does so in a way that should allow other groups to replicate the work. The novel findings, although not strictly required for a methods and resource format, are nonetheless interesting and clearly demonstrate the utility of the types of screens allowed by this approach. Therefore the work will be a valuable addition to the community.

In general, the work is well written, detailed and thorough, but there are a few blind spots that are important to address. I have provided detailed comments, but am still excited by the work and think that the concerns can be addressed with some clarification to the text, additional statistics, and a few straightforward control measurements.

1) Clarification of methods

Design of assay plates: the z-depth of the plates and thickness of gel gum are not shown. Specify whether the plates are enclosed, or whether there are air routes form one chamber to another. If the chambers are not sealed, then evaporation can deplete the concentration in the chamber in a molecule dependent manner (or travel to other chambers) which may confound the worm's response, so this is important to convey to the reader. 

2) Potential confounds

-Odor cross contamination. A control experiment is needed to show that chemotaxis in one plate is not influenced by the introduction of an oderant into a neighboring assay plate. This is especially important for volatile compounds that evaporate easily. For this control experiment, the authors should choose an oderant from their library that is among the more volatile. Similarly, residue from one set of plates could remain in the instrument across experiments. Discussion is needed of steps taken to mitigate this concern. 

-Relatedly, each small molecule likely has a different diffusion coefficient, solubility in the solvent, and absorption rate into the gum, which could impact the amount of small molecule experienced by the animal. For example, 2-nonanone, used in the library has been reported to have strong interactions with the substrate and walls: Akiko et. al., (2018). Measuring Spatiotemporal Dynamics of Odor Gradient for Small Animals by Gas Chromatography, Bio-protocol 8 (7): e2797. DOI: 10.21769/BioProtoc.2797 If the authors have an estimate for these properties they should provide them in a table. They should also comment in the text on how these properties may introduce potential confounds.

-Temperature. Consumer flatbed scanners shine intense light that heat the sample in a non-uniform way during imaging. On some scanners the light "warms up" at one end of the instrument before scanning. It appears that in this method the entire scanner is also set into an enclosed box that likely accumulates heat over the course of experiments. Because worms thermotax and are exquisite thermosensors, it is important to measure and report the spatial or temporal thermal landscapes produced by the instrument. 

o Temperature should be characterized at multiple locations within an assay plate during a typical set of experiments

o Spatiotemporal changes especially need to be measured during the 2 min scanning process 

o Temperature should also be measured across multiple scans to see how temperature fluctuates, which could introduce across-plate confounds.

3) Statistics

-The authors describe how one advantage of their approach is that it increases statistical power because it reports the individual location of each worm across plates. This allows for analysis based on the much larger number of worms (one dot per worm) rather than the number of assay plates (one dot per plate). But this approach implies that each worm is the fundamental independent measurement, and more justification is needed for that assumption. For example, there could be systematic biases shared by each assay plate: e.g. variability in the size of the small molecule droplet. One way to show that these systematics are small is to show, for a given condition, how the average worm location varies across plates (one dot per plate). Such a plot would also be valuable generally to show reproducibility across trials.

- Large screens require multiple hypothesis testing. There are now many good techniques for accounting for large numbers of hypotheses in a thoughtful way. For example, by controlling the false discovery rate:

Benjamini Y, Hochberg Y. 1995. Controlling the False Discovery Rate: A Practical and Powerful Approach to Multiple Testing. Journal of the Royal Statistical Society Series B (Methodological) 57:289-300.

Storey JD, Tibshirani R. 2003. Statistical significance for genomewide studies. Proceedings of the National Academy of Sciences 100:9440-9445. doi:10.1073/pnas.1530509100

Since this work is presenting a new type of screen, it is especially important that the authors explicitly discuss multiple hypothesis and set an example for how to interpret them in a statistically rigorous way.

4) Clarification on scientific conclusions

-Chemotaxis without capturing worms: classic chemotaxis assay captures worms around the droplet with sodium azide. Given that the arena is smaller and does not capture worms on either side, the authors should comment on how this might or might not affect the chemotaxis index. More specifically, there is a chance that preference or valence may not be consistent across 1hr. See for example this instance of valence changing on ~1hr timescale, which probably deserves a mention: Nuttley et al., 2001 https://www.ncbi.nlm.nih.gov/pmc/articles/PMC311371/

-Graph summary of odor preference: The authors should clarify if the clustering in 7B only uses valence (attraction or repulsion) or if it takes into account the magnitude. And they should clarify how this leads to the groupings in 7C. 

5) Limitations 

The "Limitations and future research" section mostly lists strengths of the method with almost no limitations. But a thoughtful discussion of limitations is warranted, including but not limited to:

o The assay ignores interactions of the small molecule with the air, solvent or substrate. So different diffusion coefficients, solubility in the solvent, and absorption rate into the gum could all result in changes to chemotaxis unrelated to the animal's intrinsic valence or attraction or repulsion to the molecule. 

o The method only reports a static snapshot in time. Therefore it doesn't capture temporal information about the animal's navigation strategy and misses phenomenon like habituation or sensitiziation that may be occurring during chemotaxis. See Nuttley paper mentioned above.

Minor:

- The equation on manuscript page 12 doesn't render properly and so I cannot see the variables. 

- The sentence about Spinosad on manuscript p. 26 is missing a main clause.

Reviewer #3: Overall, this is a very good comprehensive study including also innovative technical aspects. Two minor comments that can be easily addressed:

1)

"The primary strategy worms use to accumulate near attractants is to suppress turns (pirouettes) and to increase forward run duration when moving up a chemical gradient [6]. The converse strategy underpins the avoidance of repellents [7]."

Indeed, klinokinesis plays the central role during food seeking behaviours od C. elegans; whether it is the primary strategy is not clear and could be debated. We suggest to mention and cite cite some of the major work on klimotaxis i.e., weathervaning, as a paralell strategy.

2)

"The platform is simpl

---

## [Decision Letter · Decision Letter 2]

18 Apr 2024

Dear Dr Goodman,

Thank you for your patience while we considered your revised manuscript "An efficient behavioral screening platform classifies natural products and other chemical cues according to their chemosensory valence in C. elegans" for publication as a Methods and Resources Article at PLOS Biology. This revised version of your manuscript has been evaluated by the PLOS Biology editors, the Academic Editor and two of the original reviewers.

Based on the reviews, I am pleased to say that we are likely to accept this manuscript for publication, provided you satisfactorily address the following data and other policy-related requests that I have provided below (A-E):

(A) We would like to suggest the following modification to the title:

“A high-throughput behavioral screening platform for measuring chemotaxis in C. elegans”

(B) You may be aware of the PLOS Data Policy, which requires that all data be made available without restriction: http://journals.plos.org/plosbiology/s/data-availability. For more information, please also see this editorial: http://dx.doi.org/10.1371/journal.pbio.1001797

-Supplementary files (e.g., excel). Please ensure that all data files are uploaded as 'Supporting Information' and are invariably referred to (in the manuscript, figure legends, and the Description field when uploading your files) using the following format verbatim: S1 Data, S2 Data, etc. Multiple panels of a single or even several figures can be included as multiple sheets in one excel file that is saved using exactly the following convention: S1_Data.xlsx (using an underscore).

-Deposition in a publicly available repository. Please also provide the accession code or a reviewer link so that we may view your data before publication. 

Figure 2A-B, 2D-E, 3, 4A-C, 5, 6A-B, 7B-C

(C) Please also ensure that each of the relevant figure legends in your manuscript include information on *WHERE THE UNDERLYING DATA CAN BE FOUND*, and ensure your supplemental data file/s has a legend.

(D) As the custom code that you have generated to is important to support the conclusions of your manuscript, its deposition is required for acceptance. Please ensure that the code is sufficiently well documented and reusable, and that your Data Statement in the Editorial Manager submission system accurately describes where your code can be found. 

(E) Please ensure that your Data Statement in the submission system accurately describes where your data can be found and is in final format, as it will be published as written there. 

We expect to receive your revised manuscript within two weeks. 

*Published Peer Review History*

*Press*

Sincerely,

Richard

Richard Hodge, PhD

rhodge@plos.org

Reviewer remarks:

Reviewer #1: The authors have sufficiently addressed my comments and the comments of the other reviewers, in my opinion. Congratulations!

Reviewer #2: The revised manuscript thoughtfully addresses the previous concerns. The level of detail in response to reviewers is commendable. This is nice work that will be valuable to the community!

---

## [Editor Report · Decision Letter 3]

11 May 2024

Dear Miriam,

On behalf of my colleagues and the Academic Editor, Matthieu Louis, I am pleased to say that we can accept your manuscript for publication, provided you address any remaining formatting and reporting issues. These will be detailed in an email you should receive within 2-3 business days from our colleagues in the journal operations team; no action is required from you until then. Please note that we will not be able to formally accept your manuscript and schedule it for publication until you have completed any requested changes.

PRESS

Best wishes,

Richard 

Richard Hodge, PhD

rhodge@plos.org

PLOS
